# Overload wave-memory induces amnesia of a self-propelled particle

Maxime Hubert[1✉], Stéphane Perrard[2✉], Nicolas Vandewalle[3✉] & Matthieu Labousse [4✉]

Information storage is a key element of autonomous, out-of-equilibrium dynamics, especially for biological and synthetic active matter. In synthetic active matter however, the implementation of internal memory in self-propelled systems is often absent, limiting our understanding of memory-driven dynamics. Recently, a system comprised of a droplet generating its guiding wavefield appeared as a prime candidate for such investigations. Indeed, the wavefield, propelling the droplet, encodes information about the droplet trajectory and the amount of information can be controlled by a single scalar experimental parameter. In this work, we show numerically and experimentally that the accumulation of information in the wavefield induces the loss of time correlations, where the dynamics can then be described by a memory-less process. We rationalize the resulting statistical behavior by defining an effective temperature for the particle dynamics where the wavefield acts as a thermostat of large dimensions, and by evidencing a minimization principle of the generated wavefield.

[1] PULS Group, Institute for Theoretical Physics, Interdisciplinary center for nanostructured films (IZNF), Friedrich-Alexander-Universität Erlangen-Nürnberg, Cauerstr. 3, 91058 Erlangen, Germany. [2] Laboratoire de Physique et Mécanique des Milieux Hétérogènes, CNRS UMR 7636, ESPCI Paris et PSL Université, 10 rue Vauquelin, 75005 Paris, France. [3] GRASP, UR CESAM, Université de Liège, Allée du 6 août 19, 4000 Liège, Belgium. [4] Gulliver, CNRS UMR 7083, ESPCI Paris et PSL Université, 10 rue Vauquelin, 75005 Paris, France. ✉email: maxime.hubert@fau.de; stephane.perrard@espci.fr; nvandewalle@uliege.be; matthieu.labousse@espci.fr

Many simple active biological systems possess memory mechanisms which are commonly thought to play a key role in their statistical dynamical behaviors. However, assessing the influence of memory in such active systems is an ill-defined task[1]. Indeed, what is commonly called biological memory involves mechanisms acting at different time scales from allosteric switching ($\sim10^{-5}-10^{-3}$ s) to biochemical circuits ($\sim10^{-2}-1$ s). In contrast, in synthetic active matter, already crafting a system able to self-propel, such as light-activated colloids[2], colloidal rollers[3] or self-propelled disks[4], is by itself an experimental tour de force. As a consequence, these systems are intentionally designed to be minimalistic so there is a chance to rationalize them. Implementing a memory repository in these physical systems would raise complex experimental issues, althought, some recent robotic[5] or electronically back-controlled[6] strategies could solve this issue. Correlation times may potentially play the role for a memory of a physical system[7], but in practice, their tunabilty and controllabilty upon a variation of experimentally controllable parameters are often limited. For all these theoretical and experimental reasons, physicists, including the authors themselves, are often ill-at-ease to rationalize the influence of memory in physical or biological active systems. As a consequence, this question has often been evaded in the field of active matter[8–10], even if this multiscale and ill-posed concept has fascinated and puzzled for long[11].

In the last two decades, a candidate for such an investigation appeared with walking droplets, or walkers for short[12–14]. Experimentally, a droplet bounces periodically on a vertically and sinusoidally oscillating oil surface. The result is the emergence of a complex standing wavefield[15,16] which propels the droplet and stores information about its past positions[17–26]. Indeed, each bounce of the droplet imprints a standing long-lasting axisymmetric wave centered at the point of impact, which lifetime is controlled by the vertical acceleration magnitude of the bath. In this system, because the droplet slides down the gradient of the local liquid surface, the wavefield acts as a memory that drives the droplet motion. Crucially, the amount of information encoded in the wavefield is completely controllable in a continuous fashion through the magnitude of the acceleration of the interface.

This unique feedback between the droplet and the wavefield dynamics is at the core of a stream of research mainly motivated by analogies with quantum systems[27–39]. In addition, recent numerical, theoretical and experimental studies[40–47] have shown that the memory of the walker leads to *run-and-tumble*-like chaotic dynamics[48–50], similar to Marangoni-driven drops[51], or particles in in-silico superfluids[52]. All these studies[40–47] identify the key role of the wavefield memory in the emergent statistical behavior. Nevertheless, the dynamics and statistical properties of the wavefield are poorly understood. Indeed, most of the dynamical descriptions of the walker wavefield focus on the regime of so-called "low memory" where memory time is small[33,53], or in the infinite memory limit where correlations become irrelevant[42]. In between, in the high memory regime, the complex and highly correlated interaction between the droplet and the wavefield makes theoretical analysis a daunting task, while numerical investigations are requiring large resources to obtain statistically relevant measures.

In this article, using simulations and experiments, we show that a walker trapped in a weak harmonic potential with large memory reaches an active statistical limit where the wavefield becomes a memory-controlled thermostat. We show that an excess of memory leads to an effective memory-less particle dynamics, paving the way for further understanding of highly correlated memory-driven dynamics.

## Results

**Walker as a memory-driven agent.** A walker is the association of a sub-millimetric oil droplet, periodically bouncing on a vertically-vibrated oil surface, and the guiding standing waves generating by the drop bounces (see Fig. 1a)[17–26] (see Methods). In this article, for practical reasons, the space available to the walker is bounded by confining the particle with an applied external potential. Experimentally, the external potential is applied to the ferrofluid core of the droplet through a combination of magnetic fields originating from electric coils and a permanent magnet (see Fig. 1a and Methods). The harmonic potential per unit mass is noted $U = \omega^2|\mathbf{r}|^2/2$, with $\omega/2\pi \sim 0.01-0.4$ Hz its frequency. It is worth noticing that, since the liquid surface has no ferromagnetic susceptibility, it is not altered by the presence of the magnetic field. In addition, previous studies have focused on the different ways to confine a walker, such as circular or elliptic confinements[31,37], or quasi one-dimensional channels[34,35]. Nevertheless, these confinement strongly alter the shape of waves and add a new layer of complexity to the walker dynamics. For these reasons, in this article, the confinement is only originating from a weak external force to be as close as possible to a free walker dynamics.

The dynamics is simulated by solving the recurrent coupled equations relating the speed $\mathbf{v}$ of the walker and wavefield $\zeta$ through [Eqs. 1, 2][29]

$$\mathbf{v}(t_{N+1}) = \mathbf{v}(t_N) - \beta\tau_F\mathbf{v}(t_N) - \tau_F\nabla U(\mathbf{r}_N) - c\tau_F\nabla\zeta(\mathbf{r}_N, t_N) + \mathcal{O}(\mathbf{v}(t_N)\nabla\zeta(\mathbf{r}_N, t_N)^2), \tag{1}$$

and

$$\zeta(\mathbf{r}, t_N) = \zeta_0\sum_{p=0}^{N}J_0\left(\frac{2\pi}{\lambda_F}|\mathbf{r} - \mathbf{r}_p|\right)\exp\left(\frac{-|\mathbf{r} - \mathbf{r}_p|}{\delta}\right)\times\exp\left(-\frac{t_N - t_p}{\tau_F\mathrm{Me}}\right). \tag{2}$$

$\beta$ is the effective damping coefficient applied to the walker, $\tau_F$ is the bouncing period, $U$ is the external confining potential per unit mass, $c$ is the wave coupling coefficient, $J_0$ is the bessel function of first kind and order zero, $\lambda_F$ is the wavelength and $\delta$ is a spatial damping factor. Me is the memory parameter which controls the amount of information in the wavefield. Experimentally, Me is controlled by the temporal dampening of the wave $\tau$, controlled by the proximity of the applied vertical acceleration magnitude $\gamma_m$ of the surface to the Faraday threshold $\gamma_F$, i.e., $\mathrm{Me} = \tau/\tau_F = (1 - \gamma_m/\gamma_F)^{-1}$. One should note that several models[24,25,29,54], varying in their strategies to solve the walker's equation, have been proposed, and share the same core ingredients. They are in good qualitative agreement with the model we use. In all models, the control parameters is the amount of information, and not the duration of a single delay, in contrast with recent electronic systems with temporal feedbacks[6]. Finally, to be precise, the wave coupling slightly depends on the speed at each impact which is denoted by $\mathcal{O}(\mathbf{v}(t_N)\nabla\zeta(\mathbf{r}_N, t_N)^2)$ (see SI and Methods).

Figure 1b and c illustrate the dynamics from the waves and particle point of view. The wavefield stores information from a chain of standing-waves sources following the walker trajectory. This chain characteristic length scales as $\sim\mathrm{Me}V\tau_F$, with $V$ being the mean particle speed. We highlight that it is not possible to reduce this physical system (droplet and waves) to a point-like particle whose motion depends only on its current position and velocity. The droplet is able to read its memory (the two last terms in Eq. (1)) and to edit its memory (through the Bessel function in Eq. (2)), similarly to a Turing machine[55]. Erasing the memory is also possible with a specific protocol[55]. The concept of memory is therefore justified since the walker can therefore write, read and also edit the information it inscribes onto the liquid interface.

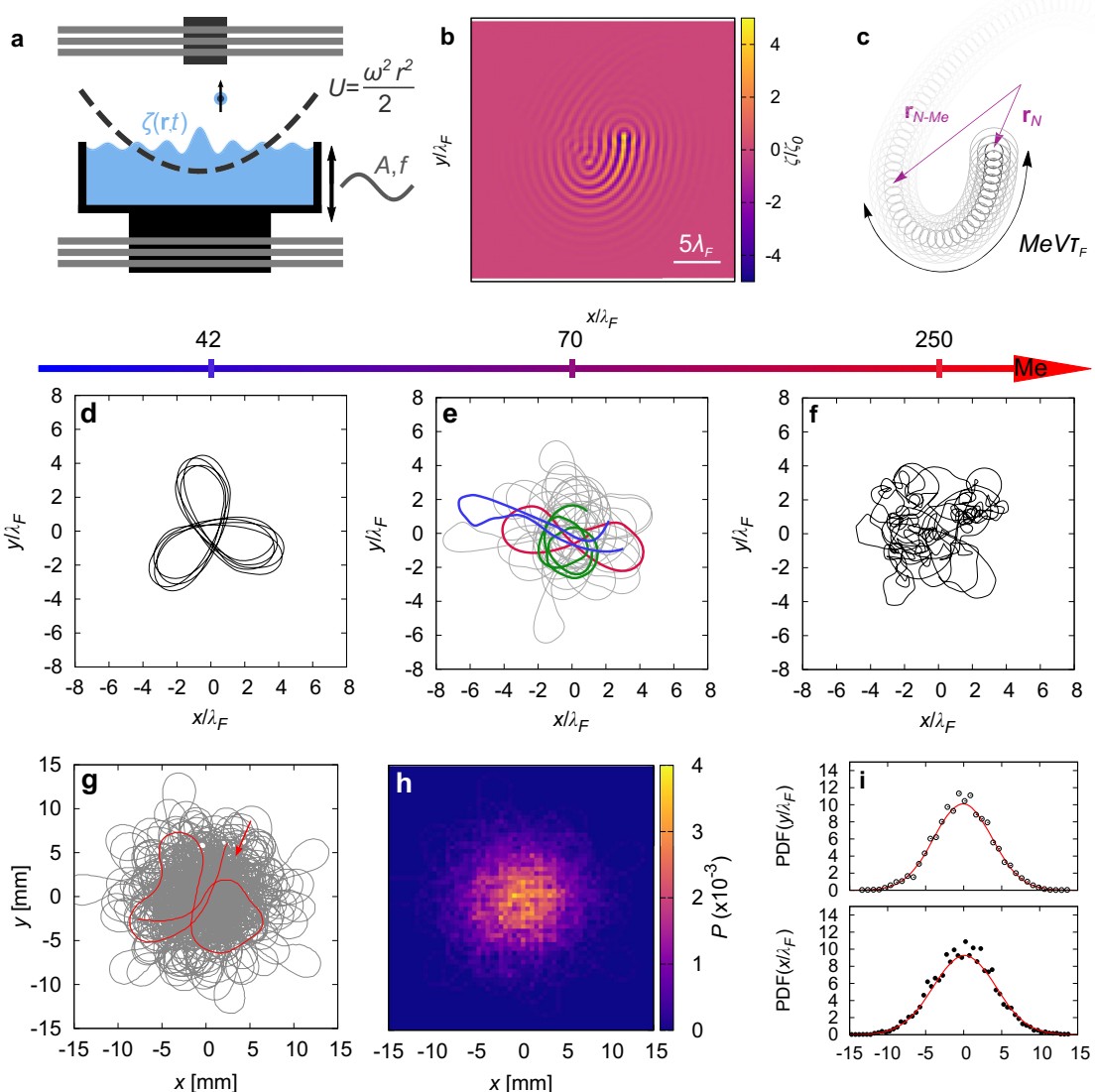

**Fig. 1 Dynamics of a walker confined in a harmonic potential in the high memory regime. a** Schematic of the simulations and experiments. A drop of silicon oil bounces onto a silicon oil/air interface oscillating vertically at a frequency *f* and an amplitude *A*. Experimentally, the core of the droplet is made of ferrofluid (see Methods). A set of coils in the Helmholtz configuration (light gray in the picture) generates a magnetic moment inside the ferrofluid core (depicted by a arrow). An additional permanent magnet (dark gray on the figure) applies a harmonic potential on the walker, confining it in the center of the experimental set up. **b**, **c** Illustration of the wavefield and associated trajectory. Decreasing gray scale is a qualitative indication of the wave sources intensity. The mean number of secondary sources contributing to the wavefield, Me, leads to a characteristic memory length, Me$V\tau_F$, with *V* being the mean speed, and $\tau_F$ being the bouncing period and the frequency of the standing waves. **d**–**f** Simulated trajectories showcasing the many dynamics of the walker when Me increases for $\omega/2\pi = 0.25$ Hz. Low memory parameters (**d**) lead to self-organized trajectory and wave sources. Medium memory parameters (**e**) generate chaotic and intermittent trajectories: green trajectories represent circle motion, red accounts for lemniscates trajectories and blue for loops. The limit of large memory (**f**) shows erratic dynamics reminiscent of the dynamics observed in[40]. **g**, **h** Experimental trajectory of a walker and the corresponding stationnary probability density function for a frequency $\omega/2\pi = 0.24$ Hz and Me = 250. **i** Probability density functions for the position *x* and *y* of the walker in the case illustrated in (**g**). The black points are the experimental distribution and the red curve is a fit by a Gaussian distribution.

**Dynamics of the particle: overview of the regimes with the memory**. In Eq. (2), the memory parameter Me acts as a control parameter which allows to numerically span several and different dynamical regimes (Fig. 1d, e and f, see also Fig. SI 2 for complementary numerical results at large Me). At low memory (typically Me < 20), only circular trajectories are observed[56] and the walker speed increases monotonously with Me. At intermediate values of the memory parameter (typically 20 < Me < 135, Fig 1d), a quantized set of close-looped trajectories are observed and result from a wave-energy minimization[33,53]. As the memory is further increased in this intermediate regime (Fig 1e), an

intermittent and chaotic dynamics is triggered where the trajectory navigates between the many possible eigenmodes of the dynamics, as investigated experimentally in[32] and theoretically in[26,42]. While a chaotic behavior has been measured and characterized in this regime, the dynamics still shows strong autocorrelation, as proved by the appearance of circle, lemniscates or loops in the trajectory. This coherence fades away as the memory is increased even further (Fig. 1f). In the high memory regime (Me > 135), the corresponding dynamics does not present any signature of patterns reported in the literature[33] and instead the dynamics shows a fully developed chaos without apparent

underlying structure which results from a Shil'nikov bifurcation[40]. In the limit Me $\to \infty$, the temporal decay of the waves is removed and each past impact contributes with the same intensity to the wavefield, which is only investigated neatly in simulations and theory[57,58]. In the intermediate, high and infinite memory regimes, the mean walker speed is relatively constant while the persistence length of the dynamics decreases with the memory parameter[40]. In what follows, we focus only on the mainly-unexplored high memory regime.

We report in Fig. 1g a typical trajectory obtained experimentally in the high memory regime. Both experimentally and numerically, the trajectories are disordered with the presence of many loopy paths without apparent underlying structure. Over a long period of time, the overall trajectory shares the same symmetry of the confining potential (Fig. 1h) and the radial probability density function is Gaussian (Fig. 1i). Interestingly, the trajectories for lower memory parameters do not lead to Gaussian probability density functions (PDF) (Fig. SI 1). In this memory regime, the self-organization of the overall trajectory (Fig. 1d) and the intermittent dynamics (Fig. 1e) leads to structured statistics, which are not Gaussian.

**High-memory dynamics of the wavefield.** The dynamical rules of the walker evolution are mediated by the information stored in a wavefield, so that the statistical properties of the trajectory are directly related to those of the wavefield through the force $\mathbf{F}_w \propto -\nabla\zeta(\mathbf{r}_N, t_N)$ (see Eq. (1)). We expand the wavefield onto a cylindrical Bessel frame of reference by using Graf's addition theorem[59] (see SI). The complex quantity

$$a_n = \zeta_0 \sum_{p=1}^{N} J_n\left(\frac{2\pi}{\lambda_F} r_p\right) \exp\left(-in\theta_p\right) \exp\left(-\frac{p}{\text{Me}}\right) \quad (3)$$

is the weight of the eigenmode invariant by a rotation of $2\pi/n$, with $n \in \mathbb{Z}$. While the simulations assume $\delta = 2.5\lambda_F$ (see Methods), the limit $\delta \to \infty$ has been used to compute the modes $a_n$ (Eq. (3)). This choice does not remove the fundamental features of the wavefield and walkers dynamics. Even though there is an infinity of eigenmodes $a_n$, only a handful number of modes are necessary, which are determined by the confinement applied to the walker. Indeed, for $n \geq 1$, $J_n(2\pi r_p/\lambda_F) \approx 0$ around $r_p = 0$, as well as all their $(n-1)$-th derivatives. Consequently the interval of $r_p$ such that $J_n(2\pi r_p/\lambda_F) \ll 1$ increases as the index $n$ increases too. As a results only a limited number of modes contributes effectively to the wave field.

Typical time series for the amplitudes $|a_n|$ are erratic as a consequence of the chaotic dynamics of the droplet[40] (Fig. 2a and Fig. SI 3a, real part of the associated eigenmodes are shown in Fig. 2b). Given the apparent lack of simple structure in their time series, we investigate their associated statistical PDF. The PDF $P(|a_n|^2)$ shows an exponential decrease (Fig. 2c), indicating that $P(|a_n|)$ follows a Gaussian distribution as would be a sum of uncorrelated memory-less events. This evolution is especially valid for large $|a_n|$. In addition, the distribution for $|a_0|$ differ from the others, presumably because this mode plays a special role since it shares the same spatial symmetry as the harmonic potential containing the walker. Note that even though the modes $n > 0$ do not share the potential symmetry, they do not vanish. This is due to the temporal decay of the waves, which lead to an axial symmetry breaking of the source positions. Furthermore, the different wave modes of amplitude $a_n$ are weakly correlated to each other (Fig. 2d, Fig. SI 3d). This correlation decreases with the memory parameter Me.

The standard deviation of $P(|a_n|^2)$ is found to be identical for $0 < n < 7$, which hints toward an equipartition of energy within the eigenmodes. Measuring the root mean squared (rms) width $a_n^{\text{rms}} = \sqrt{\langle |a_n|^2 \rangle}$ of the PDF (Fig 2e, Fig. SI 3e), we observe that

$a_n^{\text{rms}}$ decreases slowly with $n$ (barely a 10% decrease for the first 25 modes) indicating that many modes fluctuate strongly. Furthermore, the modes PDF does not change significantly with the memory parameter Me. Indeed, as Me changes from 500 to 10,000, the value of $a_0^{\text{rms}}$ changes approximately from $4\zeta_0$ to $6\zeta_0$.

We further analyze the wavefield dynamics by computing the field intensity $E$ which we defined as (see SI) [Eq. 4]

$$E = |a_0|^2 + 2\sum_{n>0} |a_n|^2. \quad (4)$$

The PDF $P(E)$ is well fitted by a Gamma distribution, especially for large Me (Fig. 2f, Fig. SI 3f, see SI for the fitting parameters). Such a choice is guided by the fact that a sum of independent variables whose distribution follows the same exponential distribution follows a Gamma distribution. Given the moderate correlations between the effectively-contributing modes $a_n$, the exponential distribution of the $|a_n^2|$ and the weak variation of $a_n^{\text{rms}}$ with $n$, this choice is expected to approximate correctly the wave-intensity distribution. Also we observe that larger Me lead to an increase and widening of $P(E)$, which is a consequence of the increasing $a_n^{\text{rms}}$ (Fig. 2e). Finally, we note that the numerical PDFs are in quantitative agreement with the experimentally-highest-reachable Me (Fig. SI 4). Three independent experiments at similar Me and different frequencies also show Gaussian $P(|a_n|)$, a slow decrease of $a_n^{\text{rms}}$ with $n$, and a Gamma-like PDF of the wavefield intensity.

To isolate the influence of the correlations between the successive walker positions, we compare the wave-driven dynamics of the walker with a random wavefield generated by the superposition of $5 \times$ Me random sources. The radial PDF of the random sources is chosen to be equal to the radial PDF of the walker position for Me $= 2500$. This random field is then statistically equivalent to the wavefield generated from randomized positions of the particle. Similarly to the case of walkers, the PDFs $P(|a_n|)$ are Gaussian (Fig. 2c, Fig. SI 3c), $a_n^{\text{rms}}$ decreases over one hundred units of $n$ (Fig. 2e, Fig. SI 3e), and $P(E)$ can be fitted by a Gamma function (Fig. 2f, Fig. SI 3f).

Yet, important differences exist. Contrarily to the walkers case, the correlation matrix shows no correlation between modes (Fig. 2d, Fig. SI 3d), a feature which results from our randomly-generated field. As Me increases, the different $a_n^{\text{rms}}$ for the random field increase by a common factor. Indeed, for a random distribution of source positions $P(r)$ with standard deviation $\sigma$, $a_n^{\text{rms}}$ reads [Eq. 5]

$$(a_n^{\text{rms}})_{\text{random}} = \frac{\zeta_0^2 \text{Me}}{2} \exp\left(-\frac{4\pi^2}{\lambda_F^2}\sigma^2\right) I_n\left(\frac{4\pi^2}{\lambda_F^2}\sigma^2\right), \quad (5)$$

where $I_n$ is the modified Bessel function of first kind of order $n$ (see SI). This result plotted in Fig. 2e implies that the number of wave modes in the case of a random distribution is only determined by the standard deviation of the distribution of sources. On the contrary, the walker case shows an increase of the number of modes storing energy while keeping a roughly identical distribution of wave sources $P(|\mathbf{r}|)$ as discussed in further details in the following sections. The increase of modes effectively contributing to the wavefield can be understood as effective degrees of freedom (DOF) which we define by [Eq. 6]

$$\text{DOF} = \frac{\sum_{n=0}^{\infty} n|a_n|^2}{\sum_{n=0}^{\infty} |a_n|^2}. \quad (6)$$

The DOF does not depend on Me for a random wavefield, while the DOF increases logarithmically with the memory in the case of a walker, as if the wavefield were acting as a reservoir of increasing dimension (Fig. 2g, Fig. SI 3g). As a conclusion, the

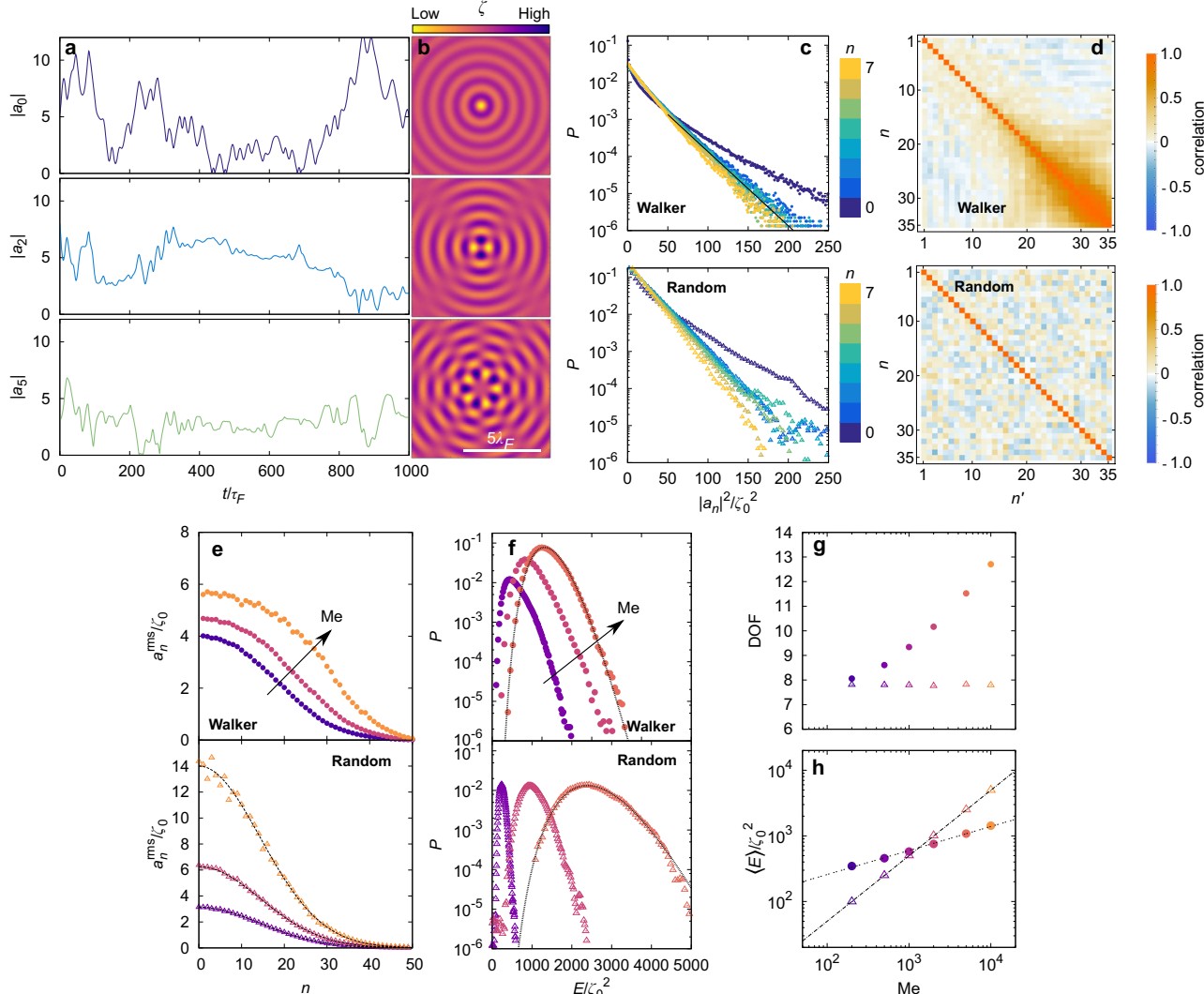

**Fig. 2 Statistical description of the numerical dynamics of the wavefield. a** Numerical time series for $|a_n|$ for several different values of $n$, for memory parameters Me = 1000 and $\omega/(2\pi) = 0.25$ Hz, and (**b**) vizualisation of the real part of the eigenmodes. **c** Numerical probability density function $P$ that the mode $a_n$ takes the value $|a_n|^2$ for a walker (resp. randomly constructed field) for $n = 0, ..., 7$ (from blue to yellow) at Me = 1000 and $\omega/2\pi = 0.25$ Hz (logarithmic scale along the $y$ axis). **d** Correlation matrix for the modes $a_n$ for $\omega/2\pi = 0.25$ Hz and Me = 1000. Both the random of the walker-generated and randomly-generated wavefield are presented. **e** Numerical root mean squared value (rms) of $|a_n|$ as a function of $n$ for memory parameters Me = 500, 2000 and 10,000 (purple, red and orange as indicated by the arrow) in the case of the walker dynamics (resp. randomly constructed field). The theoretical prediction of Eq. (5) is indicated with dotted lines. **f** Numerical probability density function for the wave intensity $E$ (in semi-logarithmic scale) for the walker dynamics (resp. randomly constructed field). Memory parameters are Me = 500, 2000 and 5000, and they follow the same color code as indicated by the arrow. **g** Numerical evolution of the number of degree of freedom DOF (Eq. (6)) as a function of the memory parameter for the walker dynamics (full circles) and the random dynamics (empty triangles) in semi-logarithmic scale along the $x$ axis. **h** Comparison of the evolution of the average wavefield intensity $\langle E \rangle$ with Me, between the walker wavefield (full circles) and the random wavefield (empty triangle) in double logarithm scale. Dashed lines are power laws fitted to the simulation data. For the walker, the exponent obtained from a power law fit and is 0.382 ± 0.024 (error is the 95%-confidence interval, coefficient of determination $R^2 = 0.9997$). For the random field, the exponent is 0.991 ± 0.007 (95%-confidence interval, $R^2 = 0.999$).

memory parameter gives a control on the properties of the reservoir surrounding the walker.

A last difference between the randomized and walker wavefield lies in the evolution of the mean wave intensity $E$ with the memory Me. The average wave intensity increases differently in the case of the walker and a random wavefield (Fig. 2h, Fig. SI 3h). The mean intensity of the random field is proportional to Me ($R^2 = 0.999$) as expected from our theoretical prediction Eq. (5), (see SI) while the correlated chain-like distribution created by walkers presents a sublinear scaling Me$^p$ with $p = 0.382 \pm 0.024$ ($R^2 = 0.999$). For a random field, the intensity per unit memory is $E/\text{Me} \sim 1$ which means that each source contributes in average equally. In contrast, the intensity per unit memory for the walker

wavefield yields $E/\text{Me} \sim \text{Me}^{-0.62}$. This decaying evolution indicates that the system tends to decrease significantly its wave intensity by selecting trajectories which promote destructive interference. A significant decrease of the wavefield intensity has already been observed for lower values of the Me with quantified trajectories[33,53]. Here we show that via destructive interference, this wave minimization mechanism extends to more complex and chaotic trajectories at high Me. Our analysis proves that the averaged modes depend not only on the probability density function but also of higher-order correlation functions, a difference with the result of Durey et al.[42] which we expect to hold when the memory time exceeds the averaging period of time required to insure ergodicity.

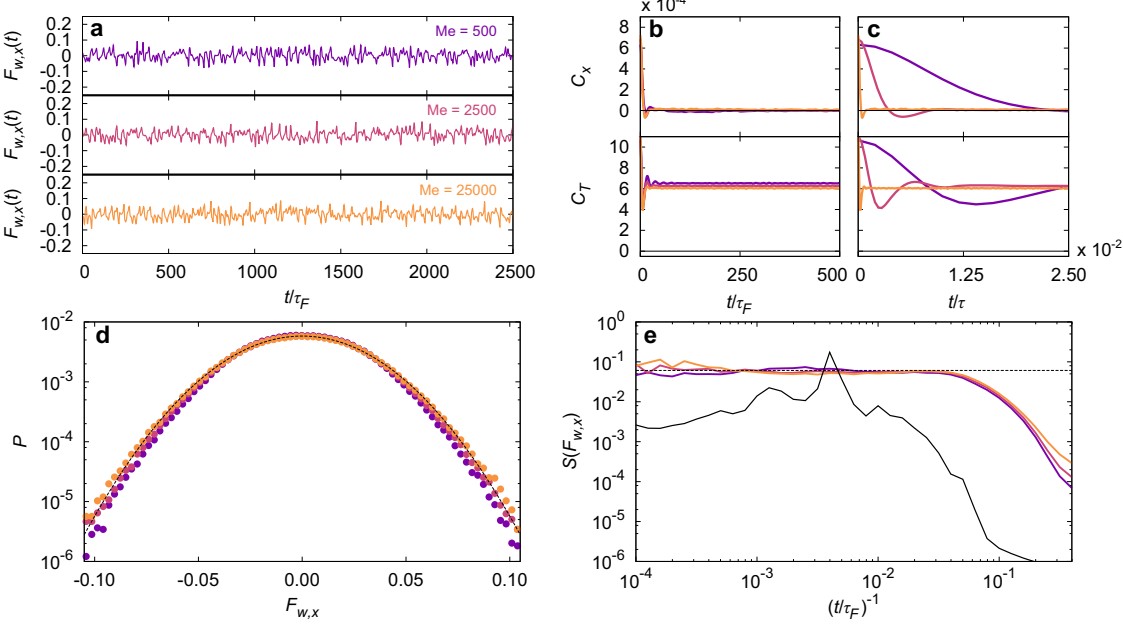

**Fig. 3 The wavefield acts as a thermal reservoir for the walker. a** Numerical time series of the wave force along the $x$ direction, $F_{w,x}$, for different values of the memory parameter Me and the same frequency $\omega/2\pi = 0.25$ Hz. From top to bottom the memory parameter is Me = 500, 2500 and 25,000 (purple, red and orange). **b,c** Numerical correlation functions for the wave force along the $x$ direction, $C_x$, (top) and along the direction tangent to the velocity, $C_T$ (bottom). The memory parameters and frequency are as in (**a**). Two points of view are presented. The time $t$ is rescaled by (**b**) the bouncing time $\tau_F$ and (**c**) by the memory time $\tau$. **d** Numerical probability density function $P$ for the wave force along the $x$ direction on a semi-logarithmic scale. Parameters are as in (**a**). The solid gray curve is a Gaussian fit to the data obtained at Me = 25,000. **e** Density power spectrum $S$ for the wave force along the $x$ direction on a double logarithmic scale. Parameters are the same as in (**a**). The dashed black line is a guide for the eyes and the black line indicates the density power spectrum in the low memory parameter regime, Me = 50.

**Markovian walker dynamics from an overload of memory**. We now consider the influence of the dimensional increase of the dynamics resulting from the additional wave DOF triggered by the walker, with increasing Me. The time series of the force $F_{w,x}(t)$ experienced by the walker solely originating from the waves along the $x$ direction is erratic, as expected from previous studies[40] (Fig. 3a, Fig. SI 5a). The $y$ direction is statistically identical given the axisymmetry of the confining potential. For the smallest memory parameter illustrated (Me = 500), we observe a correlated signal (Fig. 3b, Fig. SI 5b), especially for low frequency of the harmonic potential. This correlation exists because of the intermittent dynamics of the walker[55].

As the memory parameter Me is increased, the correlation is lost and the auto-correlation $C_x(t) = \langle F_{w,x}(t_0)F_{w,x}(t + t_0)\rangle_{t_0}$ converges toward a sharp peak at $t = 0$, which can be approximated by a Dirac function, namely $C_x(t) \simeq 2D\delta(t)$ where $D$ defines an effective diffusion coefficient. This feature is even more pronounced when renormalising the correlation time by the memory time (Fig 3c, Fig. SI 5c). Indeed, the time-correlation within $F_{w,x}$ becomes practically non-existent when compared to the amount of information stored into the wavefield for increasing Me. At memory parameter Me = 500 the correlation time is of the order of $10^{-2}\tau$, while for memory parameter Me = 25,000 the correlation time is of the order of $10^{-4}\tau$.

Along the tangential direction however, the auto-correlation for the force $F_{w,T}$ does not converge to zero. Indeed, $C_T(t) = \langle F_{w,T}(t_0)F_{w,T}(t + t_0)\rangle_{t_0}$ presents a sharp peak at the origin and a plateau at longer time which ensures the average propulsion of the particle at mean speed $v_0$. Small oscillations at high frequency are observed at short time scales, in particular at small Me. They correspond to the oscillations of the walker speed, which can trigger chaotic behaviors as described in[40]. As a consequence, the

force exerted by the wavefield on the particle can be divided into two contributions, a constant tangential component ensuring self-propulsion, and a random component.

The PDF of the force $F_{w,x}$, $P(F_{w,x})$, is Gaussian for all values of Me > 500 (Fig. 3d, Fig. SI 5d). It is worth noticing that the standard deviation of this probability density function shows a very weak increase with Me. The associated power spectral density $S(F_{w,x})$ (Fig. 3e, Fig. SI 5e) shows that for the highest memory investigated, Me = 25,000, $S(F_{w,x})$ is flat over three orders of magnitude in frequency, equivalent to a white noise. Observed deviations from the flat power spectrum have two origins. First, for smaller values of Me and low frequency (Fig. SI 5e), a small bump is observed around $t^{-1} \sim 4.10^{-3}\tau_F^{-1}$, which corresponds to the characteristic orbital period of a walker at low memory parameter[33,53,56]. This deviation vanishes for large memory parameters. The second source of deviation is observed at high frequency for all Me. It can be attributed to the non-vanishing correlations over a few ($\approx 10$) bounces as discussed in[40].

As a consequence of those observations, from the particle point-of-view, the wave reservoir eventually preserves the self-propulsion, and the fluctuating component acts as a white noise force. The wave force can be described as a combination of a simple deterministic propelling force $\mathbf{F}_p(\mathbf{v})$ aligned with the velocity which contains the correlations at short time scales, and a memory-less white noise $\eta(t)$. It is surprising to lose all correlations in a memory-driven dynamics such as it becomes approximated by a Markovian process.

**Effective temperature induced by a memory**. As a result of the previous observations, the probability $P(\mathbf{r}, \mathbf{v})$ to find the particle at a position $\mathbf{r}$ and a velocity $\mathbf{v}$ can be approximated with good accuracy by a Fokker–Planck equation[60]. Neither in the

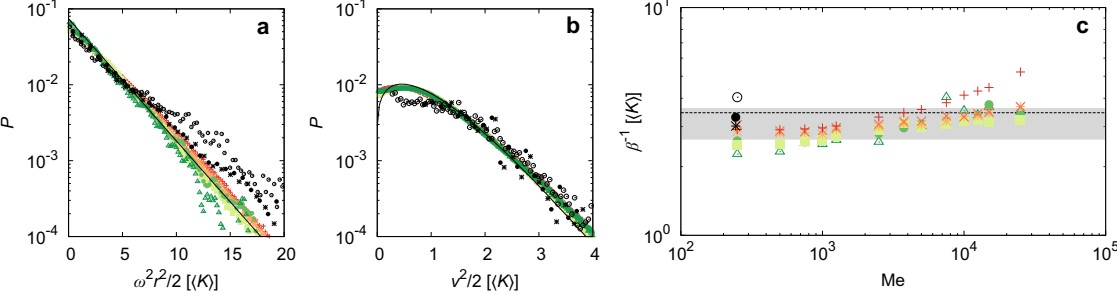

**Fig. 4 Statistical description of the walker experimental and numerical dynamics. a** Probability density function $P$ of the radial position $r$ of the walker (with respect to the center of the harmonic potential) from both simulations and experiments. The potential energy $U = \omega^2 r^2/2$ is measured in $K$ units. Colored points: simulations for a memory parameter Me = 2500 and various frequencies $\omega/2\pi = 0.25$ Hz (red plus), 0.125 Hz (orange cross), 0.1 Hz (light orange star), 0.05 Hz (yellow-green square), 0.025 Hz (light-green circle), 0.01 Hz (green triangle). Black symbols: three independent experiments at different frequencies of the harmonic potential and memory ($\omega/2\pi$ (Hz),Me): (0.20,244) (black star), (0.36,250) (black open circle), (0.30,244) (black filled circle). Black lines: theoretical predictions. **b** Probability density function of the speed $v$ of the walker from both simulations and experiments. The same color code as in (**a**) is used. **c** Evolution of the effective temperature per unit mass $\beta^{-1}$. Color code is that of (**a** and **b**) but for increasing Me parameter. Dash line: value of $\beta^{-1}$ averaged over Me. Gray area: fluctuations of $\beta^{-1}$ averaged over Me.

experimental analysis, nor in the numerical simulations, were correlations between position and velocity identified. Hence we infer the ansatz $P(\mathbf{r},\mathbf{v}) = P(\mathbf{r})P(\mathbf{v})$. For practical purposes, we define the time-average kinetic energy per unit mass $K = \frac{1}{2}\langle v^2\rangle$ which is a quantity independent of $\omega$. $K$ is also found to vary less than 0.6% with Me in the range [200:25,000]. The position PDF for the position $P(\mathbf{r})$ is well described by a Boltzmann-Gibbs probability density function $\mathcal{P}(\mathbf{r}) = \alpha\omega^2|\mathbf{r}|\exp\left(-\beta\omega^2|\mathbf{r}|^2/2\right)$ with $\alpha$ a normalization factor and $\beta^{-1}$ the equivalent temperature of our system (per unit mass) (Fig. 4a). This observation holds for both numerical simulations and experiments, and for all frequencies investigated ($\omega/2\pi \in [0.01;0.25]$) and Me > 200.

As suggested in[61], we expect the velocity PDF $P(\mathbf{v})$ to be given by $\alpha'\exp\left(-\Phi(\mathbf{v})/D\right)$, with $\mathbf{F}_p = -\nabla_v\Phi$, $D$ an effective diffusion coefficient and $\Phi(\mathbf{v})$ a velocity potential to be determined. The velocity potential $\Phi(\mathbf{v}) = \phi_0\left(|\mathbf{v}| - v_0\right)^2/2$ shows a good agreement between the theoretical prediction and the numerical speed PDF (Fig. 4b). The experimental data also collapses adequately onto the master curve. It is interesting to compare the velocity potential used here, namely $\Phi(\mathbf{v}) = \phi_0\left(|\mathbf{v}| - v_0\right)^2/2$, with the one used in previous investigations performed at lower memories[36,56] which is stiffer with the presence of $v^4$ terms. This suggest that the constrain on the self-propulsion speed $v_0$ is softer at high memory than at short memory.

However, the properties of $P(\mathbf{r})$ and $P(\mathbf{v})$ related to the walker dynamics do not change strongly with memory. This can be measured by computing the standard deviation of the PDFs shown in Fig. 4a, which also corresponds to the effective system temperature of the walker. We measure this quantity over two decades of values of the memory parameter by fitting the PDF $P(\mathbf{r})$ with a Gaussian (Fig. 4c). We obtain $\beta^{-1} \simeq 3.1 \pm 0.5 K^{-1}$. A very weak evolution with the memory Me could be argued, in which case a power law fitting $\beta^{-1} \propto Me^\nu$ gives a very small exponent $\nu = 0.096 \pm 0.021$ (95% confidence interval) when the fit is applied to all frequencies (0.010–0.25 Hz) at once to increase the precision. Mostly all the numerical and experimental data fall onto the same master curve, at the exception of the numerical points at $\omega/2\pi = 0.25$ Hz and experimental data at $\omega/2\pi = 0.2$ Hz where an inflection is observed, leading to a deviation of $\beta$ with respect to the other frequencies. This behavior might be related to the external potential which is strong and for which the chaotic dynamics are at the edge of a strongly-developed chaos. Finally, while $\beta^{-1}$ increases very weakly with Me, $\beta^{-1}$ strongly correlates

with the wave intensity $E$ linking back the features of the wavefield to the dynamics of the walker (Fig. SI 6).

## Discussion

The analysis conducted in this article reveals a complex interplay between the agent and its wavefield. If Eqs. (1) and (2) together describe a deeply non-Markovian dynamics, the build-up of the memory wavefield with increasing Me breaks the correlations in the dynamics of the walker and leads to a stochastic dynamics described in average by a white-noise-driven Markovian dynamics. In addition, the observations related to the wavefield statistical behavior allow us to conclude that the erratic nature of the droplet dynamics does not arise from a chaotic mode mixing. Indeed, as evidenced in Fig. 2d and also by Fig. SI 3d, the eigenmodes of the wavefield are still moderately correlated. Therefore, the main statistical properties cannot result from a chaotic mixing of the wave modes but rather from an increase, up to a factor seven, of wave DOF (see Fig SI 3g) which stems from the increasing memory parameters and amount of information.

We finish by discussing our results in the context of cortical waves following a thought-provoking and inspirational review article by Muller et al.[62]. Although there are differences between the two systems, and our investigation is not motivated by neuroscience, the parallel between the two systems is very intriguing. In the visual cortex, it has been shown that stimulus-evoked responses can be described by a stationary bump and a propagative wave over a domain of several millimeters. These two responses encode separately the position and the initial time of the stimulus. As a consequence, two or more stimuli generate two or more responses centered at different loci which superpose onto each other. Although, the superposition mechanism is more complex than the simple linear superposition of density waves, it has been proposed that the wave state resulting for the superposition of several cortical waves may serve as computational principles. The case of many spatio-temporally-separated stimuli, which, in our analogy, would correspond to many secondary sources, would be of particular importance. The correlation between these stimuli may be of crucial importance in information processing, as the statistical features of a field resulting either from spatio-temporally-correlated or from uncorrelated sources strongly differ.

In this article, we numerically implemented a deterministic dynamical system which stores information and showed an experimental proof of principle. In this system, the past trajectory

of the particle is encoded into a standing wavefield built by the drop previous bounces, which in return propels the particle. The unique property of this system is the control of the amount of information stored. Here, this information storage shapes the properties of the wavefield which acts as a indirect controllable thermal reservoir for the particle. In the long memory regime, we have shown that the wave reservoir conserves a short-term correlation, which is sufficient to maintain a propulsion. In the long-term limit, the wavefield possesses the properties of a white noise. The intensity stored in the waves does not diverge with memory and is self-regulated by means of destructive interference. It is very striking to observe that a system with multiple readable memories can be used to embed the properties of a thermal reservoir.

## Methods

**Experiments**. In a 14 cm-circular tank filled with 5 mm of silicon oil with viscosity $\eta = 20$ cp and surface tension $\sigma = 20.9 \times 10^{-2}$ N/m, a droplet of diameter $D = 700 \pm 50 \, \mu m$ of the same silicon oil and with a core of ferrofluid is deposited. The coalescence of the droplet in the tank is prevented by vertically and sinusoidally oscillating the tank at 80 Hz with an acceleration magnitude around $\gamma_m = 3.8g$, $g$ being the gravitational acceleration (see Fig. 1a). This leads to a droplet bouncing period which is twice the tank oscillation period, i.e., the Faraday period $\tau_F = 2/f = 2.5 \times 10^{-2}$ s and a wavelength which is the Faraday wavelength $\lambda_F = 4.75$ mm. A small amount ($\approx 5\%$ in volume) of ferrofluid (iron-cobalt nanoparticles in glycerol, magnetic susceptibility $\chi = 2.6$) is encapsulated inside the drop, so that it becomes paramagnetic. Using a combination of a homogeneous magnetic field generated by two coils in the Helmholtz configuration, and a radial gradient parallel to the bath surface generated by a permanent magnet, we confine the drop in a harmonic well, whose minimum is located at the center of the tank (see Fig. 1a). Details of the calculation for the magnetic confinement can be found in ref. [33]. The frequency of the potential well can be tuned by the distance between the permanent magnet and the oil surface. Note that the magnetic fields applied do not interfere with wave generation as the silicon oil has no magnetic properties. The magnetic force acting on the drop was calibrated by recording the motion of the drop on circular trajectories for various values of the potential well frequency. The procedure is described in details in[33]. The particle trajectory is tracked using image processing, and each drop is recorded for 1 h, corresponding to $1.5 \times 10^5 \tau_F$ and a travelled distance of $7.2 \times 10^3 \lambda_F$. We analyze the trajectory statistics for a memory parameter Me $= 250 \pm 50$, which is directly related to the magnitude of the oil surface acceleration by the formula $\text{Me} = (1 - \gamma_m/\gamma_F)^{-1} = \tau/\tau_F$, $\gamma_F$ being the threshold of the Faraday instability. The value Me $= 250 \pm 50$ corresponds to the largest memory value that we were able to reach with our experimental setup. We also focus on magnetic potential frequency $\omega/2\pi < 0.30$ Hz, allowing for large trajectories. Special care has been given to the homogeneity of the vertical vibration, by looking at the regularity and the homogeneity of the Faraday waves above the instability threshold $\gamma_F$.

**Numerical simulations**. The results presented in this article have been obtained via a discrete step algorithm[29,63]. In essence, the algorithm modelling the walking droplet dynamics consists of two phases, alternating periodically. The first phase is the "bouncing phase" where the droplet is considered as a perfectly inelastic ball, bouncing on a vertically oscillating rigid surface. If the surface oscillates with a dimensionless acceleration $\Gamma = \gamma_m/g = A(2\pi f)^2/g$, where $A$ is the amplitude and $f$ the frequency of oscillation, the dynamics of the ball is uniquely determined[64]. The dimensionless acceleration $\Gamma = 4.12$ is chosen so that the bath oscillates twice as fast as the drop. As a consequence, the relative speed at impact and the duration of contact with the interface can be computed. The former information is related to the wave intensity and therefore the amplitude of each wave, and the latter to the duration of interaction with the fluid interface. The second phase is the period during which the ball sits on the interface. During that period, the droplet get a kick of momentum from the standing wavefield in the direction normal to the liquid interface, then the wavefield is updated by adding a new wave source at the drop impact position and finally, the drop loses kinetic energy via friction with the interface. For the kick of momentum, an increase of horizontal speed is applied and reads [Eq. 7]

$$\delta\mathbf{v} = |\mathbf{V}.\mathbf{N}|\mathbf{n}, \tag{7}$$

where $\mathbf{V} = (\mathbf{v}, v_z)$ is the 3D velocity of the droplet, and $\mathbf{N} = (\mathbf{n}, n_z)$ is the 3D vector normal to the surface. Lowercase symbols corresponds to the horizontal component of the 3D vectors. The normal vector $\mathbf{N}$ reads [Eq. 8]

$$\mathbf{N} = \frac{1}{\sqrt{1 + |\nabla\zeta|^2}}(-\nabla\zeta, 1). \tag{8}$$

Following the kick of momentum, a new standing wave is created on the interface at the impact position while the droplet gets a kick of momentum in the direction

normal to the wavefield. The shape of the standing wave and its time evolution obeys Eq. (2), with $\tau_F = 2.5 \times 10^2$ s, $\lambda_F = 4.75$ mm and $\delta = 2.5\lambda_F$. Finally, the droplet speed decreases exponentially with a characteristic time scale $\tau_v = 4.5 \times 10^{-2}$ s during the contact time. The stability and reproducibility of the numerics have been tested and validated in[29,40]. The initial conditions of the walking dynamics are identical for each simulation. No waves exist on the interface prior to the particle motion. The particle starts its motion with $(x_0, y_0) = (0, \lambda/2)$ and $(v_{x,0}, v_{y,0}) = (6.66, 3.33)$ mm/s. These values were chosen to break the symmetry of the potential, while being close to the equilibrium speed value. For statistical investigations in Figs. 3 and 4, the algorithm was integrated over $5 \times 10^6$ bounces, i.e., period of the waves $\tau_F$. In order to remove transient dynamics, the first 10% of the dynamics were not considered. In Fig. 2, the algorithm was used over $2.5 \times 10^6$ bounces and 125 waves modes were considered. As previously, the first 10% of the dynamics were not considered. PDFs presented in this article are normalized such as the sum of all obtained probability equals to one.

## Data availability

All the important data sets that support the findings of our study are available at the link: https://mycore.core-cloud.net/index.php/s/bZsJUfvld9MxYbT.

## Code availability

The codes are available at the following link: https://mycore.core-cloud.net/index.php/s/bZsJUfvld9MxYbT.

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

## Acknowledgements

We would like to thank Yves Couder for past fruitful insights. The authors thank warmly Vincent Bacot and Emmanuel Fort for insightful discussions. M.H. acknowledges financial support by the Actions de Recherches Concertées (ARC) of the Belgium Wallonia-Brussels Federation under Contract No. 12-17/02. M.L. and S.P. acknowledge the financial support of the French Agence Nationale de la Recherche, through the project ANR Freeflow, LABEX WIFI (Laboratory of Excellence ANR-10-LABX-24), within the French Program Investments for the Future under reference ANR-10IDEX-0001-02 PSL. Computational resources have been provided by the Consortium des Équipements de Calcul Intensif (CÉCI), funded by the Fonds de la Recherche Scientifique de Belgique (F.R.S.-FNRS) under Grant No. 2.5020.11.

## Author contributions

M.H. performed the simulations, S.P. performed the experiments, N.V. and M.L. supervised the study. All authors participate in the redaction of the paper.

## Competing interests

The authors declare no competing interests.
