## [Peer Review File · Nature Communications]

REVIEWER COMMENTS

Reviewer #1 (Remarks to the Author):

The authors perform an experimental and numerical study of an active walker, which is constituted by a vertically oscillating bouncing drop whose active dynamics results from the self-interaction between the drop and the wavefront it generates. The paper is mainly based on nice results introduced in Ref. [26] (PRL 117, 094502 (2016)). In particular, the resulting dynamical system not only develops persistent motion, and thus the trajectories result correlated on a finite time scale as in any active system, but it is also possible to tune the information exchange between the droplet and the local liquid surface.

Understanding how information exchange might impact pattern formation is surely an important topic, however, while the paper deserves surely publication in a more specific journal, I do not see how this work could meet the Nat. Comm. criteria for the following reasons. From the point of view of the novelty, the groundbreaking mechanism behind the results presented here has been extensively discussed in [26,39-45]. The results presented here are surely interesting but incremental compared with the state of the art. Moreover, there are already active matter systems suitable for studying information exchange (see for instance Nat. Comm. 9, 3864 (2018)) and it is not clear what are the advantages of the system considered compared with the state of the art of the field. Finally, the study is performed at the level of a single walker. The extension to many walkers might be a very interesting leap that makes advances in the field, however, while the authors state the model system might be employed for information exchange between walkers, they do not provide evidence for that, no experimental nor numerical.

Moreover, if the authors decide to transfer the manuscript to a more suitable journal like Scientific Reports, I would suggest considering my comments listed below

Comments

1. I suggest to improve the description of the experimental and numerical set-up in Fig. 1-(A). How do they generate the harmonic potential in the experiment? Does the harmonic potential confine the walker? Does it impact the liquid surface?
2. I have the feeling that the concept of memory remains elusive. Usually one thinks about a bit of information that can be manipulated and stored in some way. I did not find clear how it happens and what are the consequences.
3. The message of Fig. 2-(A) is not clear. Different colors in Fig. 2-(C) and (D) refer to different n , however, this message is not clear from the figure. Moreover, the fact that "The probability distribution function for the mode $n=0$ differs from a Gaussian only for small values of $|a_0|^2$ " is not evident from

the figures. Maybe they can include a Gaussian and exponential decay as a guide to the eye. I suggest to the authors to reduce the y-axis in panel (D) from [0,16] to [0,8].

Reviewer #2 (Remarks to the Author):

Reviewer report for: Overload wave-memory induces
amnesia of a self-propelled particle

Submitted to: Nature Communications

Authors: Maxime Hubert, Stéphane Perrard, Nicolas Vandewalle,
and Matthieu Labousse

September 21, 2021

The article presents interesting considerations regarding active matter systems for which the particles are endowed with memory. I believe a sound case is made for the existence of a mechanism by which there is a transition in the ability to recover information from the wave field in a walking droplet system. The authors support their numerical findings by means of experimental studies in the parameter regime that is accessible in the laboratory and provide insight into several implications of their work for other active matter systems. I favour publication of this article, provided that some important scientific and language issues are resolved.

In what follows, I detail the issues that need attention before I can recommend the article for publication. Comments in the following sections are listed in an order that follows that of the written presentation of the text. I have made an effort to point out all the language mistakes in the text; however, I advise that the authors request further revision by another very fluent or native speaker of English who is also familiar with the
problem.

Scientific issues

1. In the first paragraph of the section titled “Walkers as memory-driven agents”, the authors claim that the waves are “self-generated”. This means that the wave generate themselves, which is not the case.

The droplet impacts generate the waves. It would be more adequate to rephrase the sentence “Walkers are the symbiotic association of a sub-millimetric oil droplet bouncing on a vertically-vibrated oil surface and a self-generated guiding standing wave” as “Walkers are the symbiotic association of a sub-millimetric oil droplet bouncing on a vertically-vibrated oil surface and the guiding waves generated by the bounces” or something along those lines.

2. Near the end of page 2, the authors state “The evolution of the system (walker and waves) depends on ...”. However, at the start of the same section, they clearly defined a walker as a new object which is the association of a droplet and the wave field it generates. Therefore, the phrase above should be rewritten as “The evolution of the system (droplet and waves) depends on ...”

3. In the last lines of page 2, the phrase “The evolution of the system [...] depends on all the previous drop impact position, and does not reduced to the motion of a single point-like particle.” needs attention. It is true that the evolution of the system does not simply depend on the location and velocity of the droplet at a given instant; however, it does only depend on the position and the velocity of the droplet plus the position and velocity of the fluid at a single time. So, on the one hand, it is not true that the evolution of the system can only be calculated if we know the whole history of the system, at least not if the system is looked at as an infinite dimensional dynamical system; and, on the other hand, the requirement for the whole history enters precisely when we want to reduce the system to a single point-like particle. That is to say, we only need the memory of the system if we wish to treat the system as reduced to a point-like location with its velocity. These lines in the manuscript would be better rephrased to “We highlight that it is not possible to reduce this physical system to a point-like particle whose motion depends only on its current location and velocity.”

4. In the first line of page 3, I suggest adding a reference to “Quasi-normal free-surface impacts, capillary rebounds and application to Faraday walkers (JFM 2019)”, which introduces a walker model different from all three there cited.

5. On page 2, a few lines before equation 3, the authors refer to the potential as being harmonic, which suggests it is of the form $U = k |r|^2$ with a fixed k ; however, they then go on to say the potential has a “vanishing stiffness” which would suggest a vanishing rather than a constant k . Moreover they give an expression for this potential (per unit mass), i.e. $U = \omega^2 r^2/2$, which has a temporal frequency (though the frequency in a harmonic oscillator would depend on the mass as well as the oscillator stiffness). So it’s unclear to me whether the term harmonic is used because the potential varies in time or because it has the shape of the potential well that is typical of a harmonic oscillator, whether ω is the natural frequency of the potential for the mass they use or their potential is time-dependent. It is also unclear why their potential is referred to as harmonic if its expression is linear rather than quadratic on distance, and what they mean by vanishing.

6. In the first paragraph of section “High-memory dynamics of the wavefield” the phrase “..., given the confinement applied to the walker,...”, suggests that only certain confinements result in infinitely many eigenmodes. A different phrasing like “..., determined by the confinement applied to the walker,...” would be better.

Language issues

1. In the abstract, the phrase “a particle propelled at a fluid surface” should read “a particle propelled on a fluid surface”.

2. The phrase “self-generated stationary waves” was likely intended to be “self-generated standing waves”.

3. In the phrase “The amount of souvenirs stored in the wave-memory field”, the word “souvenirs” is either incorrect or metaphoric. I believe the authors meant to say something like “the number of steps” or “the number of prior wave-generation instances”. Otherwise, the word should be within quotation marks. Also, “wave-memory field” literally means a field of memories of the wave kind. My guess is that they are not referring to a field of memories. I believe they are referring to the memory contained in a field of waves, in which case they meant to write the “wave-field memory”.

4. When the authors say “a throughout investigation”, I believe they meant “a thorough”. Throughout is either an adverb or a preposition, neither function fits the text, as it is used as an adjective.

5. In the first paragraph of the body of the text, the phrase “colloidal roller [3] or self-propelled disks [4], is by it-self a experimental tour de force” should read “colloidal rollers

[3] or self-propelled disks [4], is by it-self an experimental tour de force”. Also, I would be advisable to write “tour de force” in italics.

6. In the first sentence of the last paragraph in page 1, the phrase “...wave field dynamics...”, should be “wave-field dynamics”, as it refers to the dynamics of a field of waves, rather than to the wave-like dynamics of a field.

7. “All these studies [39–46] pointed out the key role of the wave-memory field in the emergent statistical behavior.” Should be corrected to “All these studies [39–46] pointed out the key role of the wave-field memory in the emergent statistical behavior.”

8. There is a missing full-stop mark (period point) at the end of the last paragraph on page 1.
9. Space before the comma should be moved to after the comma in the caption of figure 1, where it says “,Me,”.
10. In the final sentence of the first paragraph in page 2, the phrase “Experimentally, we can tuned Me up to Me 250” should read “Experimentally, we can increase Me up to Me 250”. The choice of “increase” over “tune” is just a personal suggestion (which seems adequate given that you are only providing an upper limit), however the use of present tense instead of past simple is necessary.
11. In the last paragraph of page 2, in the phrase “Indeed, the droplet is propelled along the gradient of the total wavefield”, for consistency with the rest of the text “wavefield” should be written as two separate words. Alternatively, all instances of “wave field” should be changed to a single word.
12. In the same paragraph I suggest changing “with V the mean particle speed.” to “with V being the mean particle speed.”.
13. On the title and on the first paragraph of section “High-memory dynamics of the wavefield”, “wavefield” should be two words for consistency.
14. Stil on that paragraph, eigenmode should be plural (eigenmodes).
15. In the caption of figure 2, the phrase “Memory parameters are the same as in figure and follows the same color code (D).” hould be rephrased as “Memory parameters are the same as in panel (D) and they follow the same color code.”.
16. At the end of the first column on page 5, the sentence “As the memory parameter increases σ_n^{rms} increases by a common factor” is unclear. I believe it would be better phrased “As the memory parameter increases, the different σ_n^{rms} increase by a common factor”. Increase being in the third person singular rather than plural induces one to think that this is one quantity that increases by a common factor with another (not mentioned) quantity, rather than the σ_n^{rms} being multiple quantities (one for each n) that increase in ensemble (as figure 2 suggests).

17. Below equation 7, I am not sure what the authors mean by “remote control”. Perhaps they meant to suggest that the memory provides an indirect form of control of the properties of the reservoir. It would be good if they could be more precise in their claim.

18. The title of section “Markovian walkers dynamics from an overload of memory” should lose the “s” at the end of walker as it is used as adjective to “dynamics” in this case.

19. On the last paragraph in section “Markovian walker dynamics form an overload of memory” the word “waveforce” should be two words.

20. On the second paragraph of the discussion the phrase “does not change with Me” should be “do not change with Me”.

21. On the same paragraph the phrase “another walkers” should be “other walkers”.

22. Near the end of page 8, the sentence “The intensity stores in the waves does not diverge and is self-regulated by means of destructive interference.” should be “The intensity stored in the waves does not diverge and is self-regulated by means of destructive interference.”

Language and formatting suggestions

1. When submitting a paper for review, it is ideal that the manuscript have line numbers. Otherwise, referencing the portion of the text that needs attention becomes unnecessarily cumbersome.

2. In the abstract, the sentence that starts with “Here we consider a particle propelled...” can do without the initial “Here”, as the previous sentence already said “here”.

3. In the first paragraph of the introduction, I would recommend rephrasing “at different time scales, for example, from allosteric switching...” as “at different timescales. Examples range from allosteric switching...”.

4. In the caption of figure 1, I would suggest changing the phrase “with V the mean speed” to “with V being the mean speed”.

5. In the last sentence of the first paragraph of page 2, the phrase “...of the acceleration applied to the liquid interface, above which the divergence of Me close to the Faraday threshold does not allow for proper control.” would be improved if phrased as “...of the acceleration applied to the liquid interface; above which the divergence of Me , as the Faraday threshold is approached, does not allow for a proper control.”.

6. In the last paragraph of section titled “Walkers as memory-driven agents” the sentence “We rationalize its statistical properties by starting by an analysis of the wave field.” would be better phrased as “We rationalize its statistical properties, starting from an analysis of the wave field.”.

Reviewer #3 (Remarks to the Author):

I have attached a PDF copy of my referee report.

Reviewer #4 (Remarks to the Author):

Please see attached pdf file.

Reviewer #1 (Remarks to the Author):

The authors perform an experimental and numerical study of an active walker, which is constituted by a vertically oscillating bouncing drop whose active dynamics results from the self-interaction between the drop and the wavefront it generates. The paper is mainly based on nice results introduced in Ref. [26] (PRL 117, 094502 (2016)). In particular, the resulting dynamical system not only develops persistent motion, and thus the trajectories result correlated on a finite time scale as in any active system, but it is also possible to tune the information exchange between the droplet and the local liquid surface.

Understanding how information exchange might impact pattern formation is surely an important topic, however, while the paper deserves surely publication in a more specific journal, I do not see how this work could meet the Nat. Comm. criteria for the following reasons. From the point of view of the novelty, the groundbreaking mechanism behind the results presented here has been extensively discussed in [26,39-45]. The results presented here are surely interesting but incremental compared with the state of the art. Moreover, there are already active matter systems suitable for studying information exchange (see for instance Nat. Comm. 9, 3864 (2018)) and it is not clear what are the advantages of the system considered compared with the state of the art of the field. Finally, the study is performed at the level of a single walker. The extension to many walkers might be a very interesting leap that makes advances in the field, however, while the authors state the model system might be employed for information exchange between walkers, they do not provide evidence for that, no experimental nor numerical. Moreover, if the authors decide to transfer the manuscript to a more suitable journal like Scientific Reports, I would suggest considering my comments listed below

We deeply thank the referee for the time they dedicated to our manuscript and for helping us improving our article. Given that the report addresses several issues at once, we will address each of them in the following.

- While we do agree with the referee that this surprising system has been studied over the last 15 years, the influence of the memory parameter ('Me' in the text) on the wave and the particle has only been scarcely studied as a tool for investigating the statistical physics of memory-driven dynamics. Moreover, most of the focus on the literature has been given on the 'low memory' to 'intermediate memory regime' where self-organized orbits/trajectories and low-dimensional chaos can still be observed and investigated. As a result, the high-memory regime remains elusive. Moreover, to our knowledge, investigating the properties of the high-memory regime from a thermodynamic point of view has never been done before in the context of walkers and beyond. It is our conviction that the scientific community can find interest in our research, even out of the context of this article given the very general tools and concepts developed in this article. This point is discussed in the 3rd paragraph of the introduction.
- Furthermore, focusing on the field of walking droplets, droplet-confinement is usually small and applied using solid borders to the fluid which results in a deeply altered wave-dynamics. In our article, we build on a previous publication (Hubert et al, PRE 2019) which aims at keeping the walker in the largest space possible. This aims at modelling the wavefield underneath the droplet as a monochromatic thermostat with many degrees of freedom, therefore pushing the walker in the thermodynamic limit, as discussed in the previous point. This point is discussed in the section "Walkers as memory-driven agents" §4.
- Regarding the reference raised up by the referee (Nat. Comm. 9, 3864 (2018)), while we agree that the electronic setup discussed in this article allow for interactions with the past, to our

understanding the control parameter is the delay itself and not the amount of memory with which the system can interact. Therefore, in the reference suggested by the referee, it is our understanding that the main features of our article (energy minimization in the wavefield, dimensional increase of the “wave thermostat”, appearance of a chaotic non-markovian dynamics) could be investigated with such a system. Again, the strength of our system resides in the ability to precisely control the number of “delays” in the dynamics, and not the temporal distance from a single delay. The reference suggested by the referee has been included and discussed in the introduction and below Eq. (2).

- Finally, we agree with the referee that a multi-droplet expansion would be a highly interesting extension. As a matter of fact, we started investigating the problem of wave-mediated droplet communication in numerical simulations. The idea is to have a memory-editing droplet (walker) along with a memory-reading droplet (probe). The latter cannot imprint waves on the surface and can only bounce down the gradient of the wavefield. As a result, depending on the parameters of the simulations, the memory-reading droplet can maintain a trajectory at a non-zero velocity and even reproduce the eigenstate (oval and circles trajectories) that the memory-editing droplet can generate (see Fig. Re1). Given the depth of such results, we would prefer to report these results in a forthcoming publication. However, the referee is right that this discussion is brought to the reader without introduction and might seem out-of-place. As a result, we removed the corresponding paragraph from the manuscript.

Following these comments, we adapted the introduction and “Walkers as memory-driven agents” sections of our manuscript in hope of lifting any misunderstanding regarding the goal of our manuscript and its position in the state of the art of the field. Moreover, the discussion has been revised following the referee’s comment about the multi-droplet expansion.

Comments

1. I suggest to improve the description of the experimental and numerical set-up in Fig. 1-(A). How do they generate the harmonic potential in the experiment? Does the harmonic potential confine the walker? Does it impact the liquid surface?

We thank the referee for raising this very important issue. In the experiment, the oil droplet is filled with a tiny amount of ferrofluid (approx. 5% in volume). As electric coils in the Helmholtz configuration are placed above and below the experimental setup, and since the ferrofluid core gives the droplet paramagnetic properties, the presence of a permanent magnet above the experimental setup mimics the effect of a parabolic confining potential of magnetic origin. As a result, and given the fact that silicon oil has no magnetic properties, the wave generation is not altered by the presence of the confining potential. As stated previously, this is a strength of our experimental setup, as other confinement methods (circular cavities for instance) would affect deeply the generation of waves on the surface.

To account for the referee’s comment, we improved the schematics depicting the experimental set up by adding all the details regarding the magnetic part of the experiments and expanded on the description of the experimental setup in the *method* section and in the “Walkers as memory-driven agents” section in paragraph #4.

2. I have the feeling that the concept of memory remains elusive. Usually one thinks about a bit of information that can be manipulated and stored in some way. I did not find clear how it happens and what are the consequences.

We agree with the referee and thank them for this important comment. Indeed, it is worth noticing that the walker-memory really acts as the “tape” of a Turing Machine while the walker would be the “machine” reading it. Indeed, the trajectory of the walker is decided through its interaction with the wavefield. Moreover, the walker “writes” “memories” of its own trajectory in the wave field. Additionally, it has been shown in a previous publication (Perrard et al, PRL 2016) that a shock in the oscillation of the bath can lead to a droplet locally breaking the periodic bouncing and landing later on the surface. As a result, and in some precise conditions, the walker can erase its memory, by walking back along its previous path. As a result, the memory can be written, read by the droplet itself (and others as discussed previously), erased, and most importantly stored in the wavefield, as rightfully pointed out by the referee.

To account for this comment, the section of the manuscript discussing the mathematical description of the wavefield (section “Walkers as memory-driven agents, paragraphs 1 to 3), and its dynamics has been expanded.

3. The message of Fig. 2-(A) is not clear. Different colors in Fig. 2-(C) and (D) refer to different n , however, this message is not clear from the figure. Moreover, the fact that “The probability distribution function for the mode $n=0$ differs from a Gaussian only for small values of $|a_0|^2$ ” is not evident from the figures. Maybe they can include a Gaussian and exponential decay as a guide to the eye. I suggest to the authors to reduce the y-axis in panel (D) from $[0,16]$ to $[0,8]$.

We thank the referee for their comment. Figure 2 and its counterpart in the SI have both been modified according to the referee’s suggestion. We now give in Figure 2c a color code to relate to the value of n , and we give in the new figures 2e et 2f an arrow pointing towards increasing memory parameters. We also provide an exponential fit in figure 2c to illustrate our claim. The counterpart in the SI has also been modified the same way. We sincerely believe that these changes will meet the referee’s expectations.

Figure 1 : Illustrations of the trajectories of a walker and a probe in different conditions. (Left) $Me = 10$ and $\omega/2\pi = 1.0$ Hz, (Right) $Me = 30$ and $\omega/2\pi = 0.5$ Hz

Reviewer #2 (Remarks to the Author):

The article presents interesting considerations regarding active matter systems for which the particles are endowed with memory. I believe a sound case is made for the existence of a mechanism by which there is a transition in the ability to recover information from the wave field in a walking droplet system. The authors support their numerical findings by means of experimental studies in the parameter regime that is accessible in the laboratory and provide insight into several implications of their work for other active matter systems. I favour publication of this article, provided that some important scientific and language issues are resolved.

In what follows, I detail the issues that need attention before I can recommend the article for publication. Comments in the following sections are listed in an order that follows that of the written presentation of the text. I have made an effort to point out all the language mistakes in the text; however, I advise that the authors request further revision by another very fluent or native speaker of English who is also familiar with the problem.

We thank very much the referee for the supportive and constructive comments. We also particularly thank the referee for pointing some sentences that were indeed misleading. All the comments have greatly improved the quality of the manuscript. The manuscript has been revised by a native speaker. The referee will find a detailed answer in a point-by-point fashion. We have highlighted the main changes in blue and we hope very much that the revised manuscript will meet the referee's expectations.

Scientific issues

1. In the first paragraph of the section titled "Walkers as memory-driven agents", the authors claim that the waves are "self-generated". This means that the wave generate themselves, which is not the case. The droplet impacts generate the waves. It would be more adequate to rephrase the sentence "Walkers are the symbiotic association of a sub-millimetric oil droplet bouncing on a vertically-vibrated oil surface and a self-generated guiding standing wave" as "Walkers are the symbiotic association of a sub-millimetric oil droplet bouncing on a vertically-vibrated oil surface and the guiding waves generated by the bounces" or something along those lines.

We have replaced the sentence by "Walkers are the symbiotic association of a sub-millimetric oil droplet bouncing on a vertically-vibrated oil surface and the guiding standing wave generating by the drop bounce." We have kept the word "standing" to precise the wave dynamics. Additionally, we removed any other misleading statements about "self-generation" we could find in the manuscript.

2. Near the end of page 2, the authors state "The evolution of the system (walker and waves) depends on ...". However, at the start of the same section, they clearly defined a walker as a new object which is the association of a droplet and the wave field it generates. Therefore, the phrase above should be rewritten as "The evolution of the system (droplet and waves) depends on ..."

Very true. It has been corrected.

3. In the last lines of page 2, the phrase "The evolution of the system [...] depends on all the previous drop impact position, and does not reduced to the motion of a single point-like particle." needs attention. It is true that the evolution of the system does not simply depend on the location and velocity of the droplet at a given instant; however, it does only depend on the position and the velocity of the droplet plus the position and velocity of the fluid at a single time. So, on the one hand, it is not true that the evolution of the system can only be calculated if we know the whole history of the system, at least not if the system is looked at as an infinite dimensional dynamical system; and, on the other

hand, the requirement for the whole history enters precisely when we want to reduce the system to a single point-like particle. That is to say, we only need the memory of the system if we wish to treat the system as reduced to a point-like location with its velocity.

These lines in the manuscript would be better rephrased to “We highlight that it is not possible to reduce this physical system to a point-like particle whose motion depends only on its current location and velocity.”

We agree with the referee on this point, and the sentence has been changed according to the referee’s suggestion.

4. In the first line of page 3, I suggest adding a reference to “Quasi-normal free-surface impacts, capillary rebounds and application to Faraday walkers (JFM 2019)”, which introduces a walker model different from all three there cited.

Indeed, this is a very good remark. The reference is now cited and commented.

5. On page 2, a few lines before equation 3, the authors refer to the potential as being harmonic, which suggests it is of the form $U = k |r|^2$ with a fixed k ; however, they then go on to say the potential has a “vanishing stiffness” which would suggest a vanishing rather than a constant k . Moreover they give an expression for this potential (per unit mass), i.e. $U = \omega^2 r^2/2$, which has a temporal frequency (though the frequency in a harmonic oscillator would depend on the mass as well as the oscillator stiffness). So it’s unclear to me whether the term harmonic is used because the potential varies in time or because it has the shape of the potential well that is typical of a harmonic oscillator, whether ω is the natural frequency of the potential for the mass they use or their potential is time-dependent. It is also unclear why their potential is referred to as harmonic if its expression is linear rather than quadratic on distance, and what they mean by vanishing.

We thank the referee for pointing out this poorly written part. There is indeed a typo in the definition of the U and one should read $U = \omega^2 r^2/2$ instead of $U = \omega^2 r/2$. The missing r^2 has been added. Furthermore, the experimental external potential is harmonic (as one would expect by symmetry analysis) provided the drop remain in a region not too far from the center of the potential, typically $5\lambda_F$. Beyond $5\lambda_F$, the potential is still attractive but there is a small deviation from harmonicity and weaken as the droplet moves further away. Also, the confining harmonic potential is static in time. Therefore, the use of the word “stiffness” is an abuse of language that is indeed misleading. In the previous version of the manuscript, we have implicitly used the relation $\omega^2 = k/m$, and as m is constant, we used instinctively stiffness and frequency which were not correct. This error of language has been corrected and we now only use the reference to the harmonic potential frequency. Additionally, more details about the experiments have been provided in the methods section of the manuscript, in hope of removing any remaining misunderstanding about the confining potential origin and effects.

6. In the first paragraph of section “High-memory dynamics of the wavefield” the phrase “..., given the confinement applied to the walker,...”, suggests that only certain confinements result in infinitely many eigenmodes. A different phrasing like “..., determined by the confinement applied to the walker,...” would be better.

We thank the referee for pointing out this involuntary misleading sentence. This has been revised in the latest version of the manuscript.

Language issues

1. In the abstract, the phrase “a particle propelled at a fluid surface” should read “a particle propelled on a fluid surface”.

As the abstract has been rewritten to match Nature Communication standard, this problematic sentence does not longer appear.

2. The phrase “self-generated stationary waves” was likely intended to be “self-generated standing waves”.

True. It has been corrected. Additionally, every references to “self-generated” has been removed, as discussed in the “scientific issues” #1 of this report.

3. In the phrase “The amount of souvenirs stored in the wave-memory field”, the word “souvenirs” is either incorrect or metaphoric. I believe the authors meant to say something like “the number of steps” or “the number of prior wave-generation instances”. Otherwise, the word should be within quotation marks. Also, “wave-memory field” literally means a field of memories of the wave kind. My guess is that they are not referring to a field of memories. I believe they are referring to the memory contained in a field of waves, in which case they meant to write the “wave-field memory”.

Indeed, the choice of the word “souvenir” is a literary excess from us. We have replaced it by the terms “accumulated information”. Additionally, the choice of “wave-memory field” is indeed a bit literary as well and we used “memory wavefield” instead.

4. When the authors say “a throughout investigation”, I believe they meant “a thorough”. Throughout is either an adverb or a preposition, neither function fits the text, as it is used as an adjective.

The problematic sentence has been rewritten during the resubmission process and does no longer appear in the manuscript.

5. In the first paragraph of the body of the text, the phrase “colloidal roller [3] or self-propelled disks [4], is by it-self a experimental tour de force” should read “colloidal rollers [3] or self-propelled disks [4], is by it-self an experimental tour de force”. Also, I would be advisable to write “tour de force” in italics.

Thank you. It is changed according to the referee’s corrections and suggestion in the revised version of the manuscript.

6. In the first sentence of the last paragraph in page 1, the phrase “...wave field dynamics...”, should be “wave-field dynamics”, as it refers to the dynamics of a field of waves, rather than to the wave-like dynamics of a field.

It has been corrected to “wavefield”, following comment #11 of the “language issue” part of the present referee report.

7. “All these studies [39–46] pointed out the key role of the wave-memory field in the emergent statistical behavior.” Should be corrected to “All these studies [39–46] pointed out the key role of the wave-field memory in the emergent statistical behavior.”

Indeed. It is revised. Moreover, “wave-field” has been changed to “wavefield”.

8. There is a missing full-stop mark (period point) at the end of the last paragraph on page 1.

True. It is corrected.

9. Space before the comma should be moved to after the comma in the caption of figure 1, where it says “,Me,”.

Thank you. It is corrected.

10. In the final sentence of the first paragraph in page 2, the phrase “Experimentally, we can tuned Me up to Me 250” should read “Experimentally, we can increase Me up to Me 250”. The choice of “increase” over “tune” is just a personal suggestion (which seems adequate given that you are only providing an upper limit), however the use of present tense instead of past simple is necessary.

Given the deep alteration of the manuscript to fit the format of Nature Communication, the sentence referred to by the referee is no longer present.

11. In the last paragraph of page 2, in the phrase “Indeed, the droplet is propelled along the gradient of the total wavefield”, for consistency with the rest of the text “wavefield” should be written as two separate words. Alternatively, all instances of “wave field” should be changed to a single word.

For consistency we have changed all “wave field” into “wavefield” following the referee’s suggestion.

12. In the same paragraph I suggest changing “with V the mean particle speed.” to “with V being the mean particle speed.”.

It has been changed.

13. On the title and on the first paragraph of section “High-memory dynamics of the wavefield”, “wavefield” should be two words for consistency.

We have merged all “wave + field” into a single word in the whole manuscript, as per comment #11 above.

14. Stil on that paragraph, eigenmode should be plural (eigenmodeS).

Thank you. It has been corrected.

15. In the caption of figure 2, the phrase “Memory parameters are the same as in figure and follows the same color code (D).” should be rephrased as “Memory parameters are the same as in panel (D) and they follow the same color code.”

It has been revised in the manuscript.

16. At the end of the first column on page 5, the sentence “As the memory parameter increases σ_n^{rms} increases by a common factor” is unclear. I believe it would be better phrased “As the memory parameter increases, the different σ_n^{rms} increase by a common factor”. Increase being in the third person singular rather than plural induces one to think that this is one quantity that increases by a common factor with another (not mentioned) quantity, rather than the σ_n^{rms} being multiple quantities (one for each n) that increase in ensemble (as figure 2 suggests).

Very true. It is corrected.

17. Below equation 7, I am not sure what the authors mean by “remote control”. Perhaps they meant to suggest that the memory provides an indirect form of control of the properties of the reservoir. It would be good if they could be more precise in their claim.

The word “remote” is not useful so it has been removed. Following this comment, the terms “remotely controllable” in the conclusion have been removed and changed to “Indirectly controllable” which we hope follows the referee’s suggestion.

18. The title of section “Markovian walkers dynamics from an overload of memory” should lose the “s” at the end of walker as it is used as adjective to “dynamics” in this case.

Correct. It has been revised.

19. On the last paragraph in section “Markovian walker dynamics form an overload of memory” the word “waveforce” should be two words.

The multiple instances have been corrected

20. On the second paragraph of the discussion the phrase “does not change with Me” should be “do not change with Me”.

The sentence does no longer appear in the new version of the manuscript.

21. On the same paragraph the phrase “another walkers” should be “other walkers”.

The sentence does no longer appear in the new version of the manuscript.

22. Near the end of page 8, the sentence “The intensity stores in the waves does not diverge and is self-regulated by means of destructive interference.” should be “The intensity stored in the waves does not diverge and is self-regulated by means of destructive interference.”

Correct. It has been revised.

Language and formatting suggestions

1. When submitting a paper for review, it is ideal that the manuscript have line numbers. Otherwise, referencing the portion of the text that needs attention becomes unnecessarily cumbersome.

The lines are numbered now.

2. In the abstract, the sentence that starts with “Here we consider a particle propelled...” can do without the initial “Here”, as the previous sentence already said “here”.

To fulfil the stylistic requirements for publication to Nature Communications, we had to remove from the manuscript the sentence pointed out by the referee.

3. In the first paragraph of the introduction, I would recommend rephrasing “at different time scales, for example, from allosteric switching...” as “at different timescales. Examples range from allosteric switching...”.

Thank you for the suggestion. It has been changed.

4. In the caption of figure 1, I would suggest changing the phrase “with V the mean speed” to “with V being the mean speed”.

It has been changed.

5. In the last sentence of the first paragraph of page 2, the phrase “...of the acceleration applied to the liquid interface, above which the divergence of Me close to the Faraday threshold does not allow for proper control.” would be improved if phrased as “...of the acceleration applied to the liquid interface;

above which the divergence of M_e , as the Faraday threshold is approached, does not allow for a proper control.”

Thank you for the suggestion. We have revised the manuscript accordingly.

6. In the last paragraph of section titled “Walkers as memory-driven agents” the sentence “We rationalize its statistical properties by starting by an analysis of the wave field.” would be better phrased as “We rationalize its statistical properties, starting from an analysis of the wave field.”.

The sentence does no longer appear in the new version of the manuscript.

We would like to thank again the referee for their comments and the careful review of the manuscript which significantly help in improving its quality.

Reviewer #3:

The authors present the results of a combined experimental and theoretical investigation into the dynamics of a memory-driven, wave-propelled particle, whose motion is spatially confined by a harmonic potential. The distinguishing features of this fluid system are the tunable duration of the droplet's path memory and the generation of the accompanying wave field, which acts as a dynamic potential. In a very interesting series of experiments and numerical simulations, the authors demonstrate that, contrary to what one might anticipate, increasing the droplet's path memory actually leads to a decrease in long-range time correlations. Furthermore, the authors consider the statistical properties of the wave field in the high-memory limit, demonstrating that the wave field possesses properties of a white noise.

I find the results of the article surprising and fairly comprehensive, and I believe the work will be of interest to both the active matter community and the broad readership of Nature Communications. I have listed below some ideas on how to make the story more compelling, particularly for a non-expert reader. While the quality of writing is generally reasonable, there are many small writing errors that disrupt the flow of the manuscript; an extensive, but non-exhaustive, list of writing errors is included below. The mathematical content is broadly correct, although some aspects could be tightened (without changing the main results), particularly with regards to defining notation. The figures are very clear, but I have suggested some minor improvements. Finally, the authors have included a helpful reference to a repository containing an implementation of the numerical algorithm used in the paper, allowing for the simulation results to be reproducible.

Summary: The authors should address the following comments before I can recommend publication.

We thank very much the referee for the supportive and constructive comments, especially to make the story more compelling for the broad readership of Nature Communications. We also particularly thank the referee for pointing some sentences that were indeed misleading. All the comments have greatly improved the quality of the manuscript. The referee will find a detailed answer to the general and detailed comment in a point-by-point fashion. We have highlighted the main changes in blue in the main manuscript and in the SI, and we hope very much that the revised manuscript will meet the referee's expectations.

General comments

1. A cynical reader not familiar with the walking droplets literature might be entirely unsurprised that the statistical distribution in the high-memory limit is of Gaussian form – indeed, a Gaussian is what most readers would guess. However, someone in the walking-droplets community would find this result surprising, as the properties of the system (memory and monochromatic wave field) conspire to give intermittent chaotic dynamics at intermediate memories, resulting in structured statistics (as detailed by S. Perrard and M. Labousse in their prior work). In order to make a more compelling story, further background information into the intermittent dynamics and statistics of the system at intermediate memory should be included, with an additional figure (either in the main text or supplementary material). That way, the story is that intermediate memory leads to structured statistics, whereas excessive memory leads to unstructured (Gaussian) statistics. I do not expect the authors to discuss the quantum analogs detailed in prior work, but I do think that the authors are underselling their new results by not putting their work in proper context.

We thank the referee for their valuable suggestion. For any reader to understand the reach of our investigation, we provide in the new figure 1 trajectories illustrating the dynamics of a walker stuck in an eigenmode of the harmonic potential (a trifolium in this case) and also intermittent trajectory for higher memory parameters. Additionally, we provide in the SI, with the new Fig. SI1, the probability distribution functions of the walker positions for the trajectories shown in the main Fig. 1. As pointed

out by the referee and now illustrated in our manuscript, the positional PDFs do not have a Gaussian structure at low and intermediate memory but become Gaussian for high memory parameters. Finally, a short discussion has been added in the third paragraph of the second column of page 2.

2. The second-most surprising result is the loss of long-time correlations in the droplet trajectory at high memory. Currently, this result is buried in small (and rather uninspiring) panels in figure 3b). I think the manuscript could be improved by giving this result greater prominence, both in the main text and in the figures. Perhaps the authors could include a schematic-like figure demonstrating the relative memory length and correlation length of the droplet trajectory. At intermediate memory, I anticipate that these lengths are comparable, yet at higher memory the memory length becomes longer and the correlation length decreases (according to the results).

Again, we would like to thank the referee for their valuable comments, and we can only agree with their suggestions. We changed figure 3B in the new version of the manuscript to add another panel (new Fig 3C). In this new figure, the time is scaled by the memory time τ instead of the Faraday period τ_F . As a result, and as suggested by the referee, the correlation function which displays a significant time correlation for memory parameters of the order of 10^2 shows a convergence to a Dirac delta for memory parameters of the order of 10^4 . Additionally, a description of the figure has been added in the main text, in the second paragraph of the section “Markovian walker dynamics from an overload of memory”. We sincerely hope that the referee will be satisfied by the change implemented.

3. The authors demonstrate that the statistical properties of the wave field are the same as that of a randomly constructed field, with wave sources randomly distributed according to the (radial) probability density function of the droplet position. To me, this result appears very similar to the theoretical result of Durey, Milewski & Bush (Chaos, 2018), linking the droplet probability density function (assuming ergodicity) to the average wave field. Although the authors of that work presented the results in physical space (taking the form of a convolution), the statistical properties of each Bessel mode can be found in the proof of Theorem 1. If the result presented by the present authors is different from that of Durey et al., then some contrasting statements should be included. If the results are the same, then the authors should utilize that result, truncating the associated section of the supplementary material.

The referee raises here an important point, which raises some critical issues with our manuscript. Through the comparison between the randomly-created wave field and the walker-generated field, we want to highlight the significant differences between the two systems. While the two systems show the same $|a_n|$ distributions, and the same wave energy distributions, differences appear in the values of $a_{rms,n}$ (only the results for the random field can be fitted by Eq.(5)), in the evolution of the DOF with the memory, in the evolution of the mean field energy with the memory and finally, as highlighted by our new figure 2D in the cross correlation between wave mode. Indeed, for the latter point, correlations between the position of each wave sources lead to a small but non-zero correlation between neighboring modes, a feature missing in the randomly-generated field.

As a result, the two systems are **not** the same, and this is due to the correlation between the wave sources. Consequently, our findings do not follow the theory proposed by Durey et al. Nevertheless, we expect our findings to match theirs in the limit of infinite memory parameters. Indeed, in such a case, despite the walker’s wave sources aligning along the walker’s trajectory, the absence of time decay of the source wave amplitude breaks the correlations between successive sources. Therefore, the walker trajectory appears as random, and the two systems coincide.

We added an additional discussion in the main manuscript to avoid any misunderstanding and strengthen our conclusions related to the comparison between the two systems. Modifications are at the end of the “High-memory dynamics of the wavefield” section.

Detailed comments

1. Page 2, figure 1b): It would help the reader if the domain were smaller so that the details of the wave field (i.e. the novelty of the system) were more evident. If it is possible to provide a higher resolution plot, then that would help too.

We thank the referee for their very good suggestion. It has improved in the current version of the manuscript in the new version of Figure 1. We hope the referee will be satisfied of the change.

2. Page 2, figure 1c): Perhaps solid circles would be easier on the eyes than dashed circles?

We improved the graphical quality of the new fig 1c according to the referee's suggestion. We hope the changes meets the referee's expectation.

3. Page 2, figure 1d): It might help the reader if directional arrows were added to the red portion of the trajectory.

Thank you for this suggestion. The new figure 1 has been changed accordingly.

4. Page 2, figure 1e): A radial cut of the histogram might help the reader understand the statistical form, but there might not be space on the figure.

Following the referee's suggestions, we added in the new Figure 1I two panels which aims at illustrating the gaussian behavior of the distribution. A cut along the x and y axis are provided, along with a fit with a gaussian PDF. The associated description has been added in the main text and caption of the figure.

5. Page 2: The system would be more clear if the notion of bath acceleration was introduced sooner. For example, the authors previously state (page 1) 'the amount of information encoded in the memory field is completely controllable in a continuous fashion through the interface acceleration amplitude'. A reader unfamiliar with Faraday waves might wonder how one might control the acceleration of an interface (which seems very challenging without background knowledge). A couple of sentences in the introduction describing the applied vibrational forcing would surely help the unfamiliar reader.

This is a very good suggestion. We have added a sentence in the second paragraph of the introduction discussing how the acceleration is controlled in experiments, namely through the bath vertical amplitude of oscillation and frequency of oscillation.

6. Page 3, equation (4): The authors appear to be implicitly neglecting the exponential spatial decay of the wave field. A sentence explaining that the fundamental features of the wave field can be understood by considering the analytically tractable limit of $\delta \rightarrow \infty$ would make this step more transparent.

The referee is perfectly right. This has been specified just after Eq. (3) in the new version of the article.

7. Page 3, '...only the first few tens of modes are to be considered': Presumably the authors can justify the range of modes considered by noting that the harmonic potential, in effect, prescribes a maximum distance that the droplet can ever be from the origin. Then one can use the shape of the high-order Bessel functions (J_n for large n) to appropriately truncate. At the moment, the reader might be left wondering why one can only consider certain modes.

The referee is right, and to account for that lack of clarity we added a new description in the second paragraph below Eq.(3) that discuss the properties of the Bessel functions and why they can be neglected above a given value of n related to the confinement of the droplet.

8. Page 3, and elsewhere in the manuscript: The authors refer to a probability distribution, e.g. $P(|a_n|^2)$. However, I believe it would be more precise to refer to P as a probability density function (or an approximation thereof).

It is indeed a probability density function. It has been revised everywhere in the manuscript for all occurrences.

9. Page 3, ‘...differs from a Gaussian only for small values of $|a_0|^2$ ’ and ‘...the probability distributions are Gaussian...’: The authors do not provide sufficiently clear evidence that P is Gaussian-like. Perhaps a characteristic Gaussian distribution can be included on figure 2c). The authors must provide more complete evidence of Gaussian profiles.

Thank you for this comment. Figure 2c is a lin-log representation of P as function of $|a_n|^2$. Excepted for the mode 0, the curves are affine with negative slope which means $\text{Ln}P = -A*|a_n|^2+B$ with A positive. This is equivalent to $P=\text{Constant}*\exp(-A*|a_n|^2)$ which is a normal distribution for $|a_n|$ and an exponential one for $|a_n|^2$. We have added a fit to emphasize this result. To lift any confusion, we also rewrote the section of manuscript describing Figure 2c. Changes can be found in the third paragraph of Section “High-memory dynamics of the wavefield”.

10. Page 4, figure 2c): It would help the reader to immediately understand the different colors if an arrow indicating the general direction of increasing n were included. Likewise for increasing memory in figure 2d).

Indeed, thank you very much for the suggestion. Given the fact that all colors mix together, we instead added a color code on the side of the graph to help understand the graph. It has been added in the new Figure 2 and in the new Figure SI 3.

11. Page 4, ‘The standard deviation of the $P(|a_n|^2)$ is found to be identical for all the first modes...’: The authors should specify more clearly what they mean by ‘all the first modes’.

It concerns the modes $0 < n < 7$. It has been specified in the revised manuscript, in the second paragraph of the second column on page 4.

12. Page 5 and figure 2e): Is there anything significant about the choice of fitting the data to a Gamma distribution? If so, it would be good to expand upon this fit.

At this stage, this is no more than an educated guess. Our sole justification to use such a function lies in the fact that the sum of random independent variables distributed according to the same exponential distribution follows a Gamma distribution. Given that the $|a_n|^2$ follow similar exponential distributions for low values of n , we can expect the gamma distribution to provide a good approximation of the real PDF of the wave intensity. To give more information about the fitting procedure, we provide in this report the fitting formula along with the fitting parameters:

$$f(x) = \frac{1}{c \Gamma(a) b^a} x^{a-1} e^{-\frac{x}{b}}$$

And the parameters are

	Walker 100mHz	Walker 250mHz	Random 100mHz	Random 250mHz
a	12.101	18.456	41.61	17.84
b	197.421	60.194	58.48	141.86
c	100.806	49.447	18.50	19.50

We have added the explanation above in the main text in the hope to lift any misunderstanding. Changes can be found in the fourth paragraph of Section “High-memory dynamics of the wavefield”. Yet, since the parameters do not carry any physical insights, we did not include them in the main article.

13. Page 6, ‘Not only is the walker navigating space in a way that minimizes the intensity stored...’: I was surprised that the authors did not connect this result to the prior work of S. Perrard and M. Labousse, namely references [32] and [33]. The authors have included further evidence to support a very important result.

We thank the referee to point this connection which is indeed not obvious, and which strengthen our message.

14. Page 8: This is a minor point, by why does the frequency of the harmonic potential used in experiments differ to those considered in the simulations?

The experimental frequency is ≈ 0.24 Hz with an uncertainty that is hard to evaluate experimentally (likely few 0.01 Hz). We wanted to compare our numerical simulation with the experimental results. We have simply chosen among our numerical data the ones which were the closest: 0.25Hz. We present also numerical result for 0.1Hz to show the broad range of application of our findings and not restrict them to comparison with our experiments.

15. Supplementary material, page 4: One may transition directly from equation (SI 5) to (SI 10) by applying Parseval’s identity in the azimuthal direction. Currently, the authors are re-deriving this well-known identity without giving credit. The authors should cite Parseval’s identity and consider condensing this section of the derivation.

The referee is perfectly right when saying that going from SI5 to SI10 can be realized by using Parseval’s identity in the azimuthal direction. We have added a sentence about the use of Parseval’s identity and shorten the calculations and change the calculation done in the SI accordingly. We hope that the referee will be satisfied by the changes done.

16. Supplementary material, page 5: Given that E is a divergent integral (specified later), one cannot define equation (SI 5). The definition of wave field intensity should be used here instead. It is also confusing that different definitions of E (in terms of dimensions) are given in the main text and supplementary material. Given these points, and also the mis-calculation (or mis-writing) of the integral in equation (SI 12) (discussed in the next point), the authors should instead define

$$E = \lim_{R \rightarrow +\infty} \frac{k_F}{2R\zeta_0^2} \int_0^R \int_0^{2\pi} \zeta^2(r, \theta) r dr d\theta$$

which is dimensionless. By Parseval’s identity, it follows from equation (SI 10) that

$$E = 2\pi \sum_{p=-\infty}^{\infty} |a_p|^2 \left[\lim_{R \rightarrow +\infty} \frac{k_F}{2R} \int_0^R J_p^2(r, \theta) r dr \right]$$

When coupling with the corrected integral result below, one obtains

$$E = 2\pi \sum_{p=-\infty}^{\infty} |a_p|^2 \frac{k_F}{2} \frac{1}{k_F \pi}, \text{ giving } E = \sum_{p=-\infty}^{\infty} |a_p|^2$$

This result is now consistent with the equation stated in the main text.

We would like to thank the referee for suggesting this modification and for verifying the validity of our calculations. Indeed, it is more rigorous to fix the upper limit of the integral and then take the limit. It has been revised in the current version of the manuscript. Additionally, we revised that section of the SI to match the notation and definition of the main manuscript. Overall we hope that this part of our work is more precise and that the referee will be satisfied of the changes done.

17. Supplementary material, page 5: The integral in (SI 12) is incorrect, as can be spotted immediately by noticing that the dimensions do not match (the left-hand side has units of length, whereas the right-hand side is dimensionless). To correct this result, it is readily verified that

$$\frac{2}{R^2} \int_0^R r J_n^2(k_F r) dr = [J_n'(k_F R)]^2 + \left(1 - \frac{n^2}{(k_F R)^2}\right) J_n^2(k_F R)$$

Using that $J_n^2(x) + [J_n'(x)]^2 \sim 2/(\pi x)$ as $x \rightarrow \infty$, it follows that

$$\lim_{R \rightarrow +\infty} \frac{1}{R} \int_0^R r J_n^2(k_F r) dr = \frac{1}{k_F}$$

correcting the result stated in equation (SI 12), the difference here being the factor of $1/k_F$.

Again, we thank the referee for pointing out this mistake, and again for verifying our calculation. We have corrected the SI accordingly with the formula

$$\lim_{R \rightarrow +\infty} \frac{1}{R} \int_0^R r J_n^2(k_F r) dr = \frac{1}{\pi k_F}$$

Comments pertaining to notation

We have not responded to these particular comments in a point-by-point fashion as kindly proposed by the referee but have directly implemented the suggestions. The systems of notation have been completely checked. We thank the referee for spotting all these imprecisions and some internal notation inconsistency. About the specific point 19, the probability distribution is a normal distribution for x and y . Each have a prefactor $1/\sqrt{2\pi\sigma^2}$ which gives a $1/(2\pi\sigma^2)$ prefactor when you multiply them.

Comments pertaining to writing

We have corrected all spelling and grammatic mistakes. The manuscript has been checked by a native speaker. We tried to do our best to keep the British English. We have implemented almost all the suggestions. We have only kept the suggestion one as is as it will not lead to any misunderstanding nor ambiguity.

Comments pertaining to the References section

We have corrected all the reference mistakes pointed out by the referee. We thank the referee for her/his careful check.

Reviewer #4:

This article concerns the self-propelled bouncing droplet, which is a wave-particle association with many interesting physical properties. The authors study the information stored in the wavefield. Couder and Fort, and their collaborators, have shown that the droplet's previous bounces play a role in the wavefield memory. A memory parameter has been used in many studies (see below (3)). It has connections with the strength of the (Faraday) forcing, namely the vibration of the underlying fluid bath. In the present study the authors consider the high memory regime, in the sense of strong forcing near the Faraday threshold. As a consequence the particle (droplet) is under a chaotic dynamics. The main conclusion of this work is that in this high memory chaotic regime, the "overload of wave-memory induces amnesia, namely the loss of memory in the wavefield.

The problem is interesting. The arguments leading to the final conclusion are not easy to follow and therefore to be fully accepted. I would summarize my doubts in a single question: **is it really the excess of memory (through the many wave-associated degrees-of-freedom) that produces (ultimately) the Markovian process or is it the chaotic dynamics altogether exploring as much as possible the configuration space?** In other words, this work seems to provide evidence of the wave-particle dynamics being mixing, in the Dynamical Systems/Probabilistic terminology. A system is mixing if the states are asymptotically independent, namely as time evolves the measurements at different times become independent. This appears to be the case here, but it needs to be clarified and put into perspective with known dynamical systems' properties such as mixing and what is really novel about the dynamics. We reproduce some of the authors' remarks which corroborates with the need of clarifications. For example, the authors' remark (on page 6) that it is surprising to lose all correlations in a memory-driven dynamics such as it becomes approximated by a Markovian process. Furthermore in the DISCUSSION, page 8, If Eqs.(2) and (3) together describe a deeply non-Markovian dynamics, the build-up of the memory field through the parameter M_e breaks the correlations in the dynamics of the walker and leads to a stochastic dynamics described in average by a white noise-driven Markovian dynamics. Again, **high memory (forcing) leads to chaotic dynamics. But is it indeed the excess of memory (of wave-associated degrees-of-freedom), or the chaotic (mixing) dynamics, that leads to the breakdown of correlations?**

We would like to thank the referee for his very constructive remarks. The difference between dimensions mixing and time-delay effects (leading to an excess of wave-associated degrees-of-freedom) is indeed very important. We have added additional discussions in the text as well as two supplementary figures (2d and SI3d) to clarify this crucial aspect that we summarize as follows:

While from a "dynamical systems theory" point of view the system dimension is infinite, in practice however, only a moderate number of dimensions plays a role (≈ 8 to 13 degrees of freedom for the wave field and 4 for the drops (position and velocity) in the case $f = 0.25$ Hz, 16 to 27 degrees of freedom for the wave field in the case $f = 0.10$ Hz). Even if it was not the central point of these previous investigations, this moderate number of dimensions has been observed at much lower memory (e.g. in Perrard et al PRL. (2014)). The new figures 2d and SI3 in the revised version of the manuscript shows the cross-correlation between wavemodes of different indices n . The time of observation is of 50000 bounces which corresponds to 10 times the memory length (i.e. 50 times the memory parameter). As it can be seen, a correlation is still observed in the case of walker (in contrast with the randomly created wavefield). **From that perspective, the modes are still correlated in the high memory regime. As a consequence the main statistical properties cannot result from a chaotic mixing of the wave mode but rather than from an excess of "wave degrees of freedom"**. This is the key point of our article, the chaotic-like aspect of the droplet dynamics (as evidenced by Figures 3 and SI 5) is rooted in the

construction and dimensional expansion of the wavefield (as evidenced by Figures 2d, 2g, S13d and S13g). Discussions about that point can be found in the “discussion” section, paragraph 3.

Finally, I do not understand the following remark, in the CONCLUSION: This wave self-organization illustrates a novel mechanism that defines a strategy to process and analyse spatiotemporal vectorial information.

The wave field is a function of the relative positions of the previous bounces. So the amplitude and phase of the wave modes truly depend on vectors and not scalars. We fully agree with the referee that the sentence was not clear and not useful. We have suppressed it accordingly.

Typos that were found:

The manuscript has been checked by a native speaker.

(a) page 2, column 2: we can tuned

We thank the referee for pointing out this error. It has been corrected.

(b) page 5, column 1: detailed in Supplementary

It has been changed in the revised manuscript.

(c) page 8, column 1: the data numerical and experimental data fall...

It has been corrected.

(d) page 8, column 2: several typos as cortex, is HAS been (...) several millimeterS long (...) parRalell between cortical (...) spatiotemporally-sepArated (...) intensity storeD in the waves.

Thank you very much for carefully proof-reading our manuscript. These issues have been revised in the new version.

REVIEWERS' COMMENTS

Reviewer #1 (Remarks to the Author):

I would like to thank the authors for having considered my comments. The answers are clear and the improvement of the manuscript has definitely strengthened the work. From my side, the answers of the authors made clear the difference between this work and the previous ones based on the same techniques and recent studies in Active Matter. I think their improvements made the revised version of the manuscript suitable for Nat Comm. after a minor revision.

Minor points

1. Maybe the question is silly, but I think it might naturally emerge for a reader that is not an expert in the specific field: Why the regime between low and infinite is called "high memory regime"? This definition makes infinite small than higher and thus sounds odd. I would ask the authors to explain better these three regimes.

2. Fig 1 In the caption the definition of τ_F might help.

3. Eq. (1): I found "higher-order term" too generic

4. It is still not clear, at least to me, the link between τ_F and the control parameters of active particles, e.g. self-propulsion speed, persistence length, etc... For instance: is there any relationship between memory length and persistence length of the particle?

Reviewer #2 (Remarks to the Author):

The authors have attended all my requests, so I have no further points to raise. Some aspects of the work escape my expertise, but I trust these are well covered by the other referees comments.

Reviewer #3 (Remarks to the Author):

The authors have made careful and comprehensive changes to the manuscript, amply satisfying my initial queries. I am now happy to recommendation publication of the manuscript in Nature Communications.

Reviewer #4 (Remarks to the Author):

The main question I have asked is a difficult one and the authors have made an attempt to address it. In the rebuttal the authors mention about the moderate number of dimensions (that can go up to 27 DOF for the wave field). And just below, that the main statistical consequence is from the excess of wave DOF. This is an example where the claims don't quite help. I find the reading still hard to follow, at least from my perspective.

In the Discussion, the authors have added (*):

"As a result, the chaotic-like dynamics of the walker comes from the synergistic interactions between the particle and the wavefield, a feature which stems from the increasing memory parameters and amount of information." That the dynamics of the walker comes from the synergistic interactions between the particle and the wavefield, is the known vital ingredient of this wave-particle association. The amnesia-effect that stems from the increasing amount of information still puzzles me.

(*) In the rebuttal the authors write "Discussions about that point can be found in the "discussion" section, paragraph 3." I found some discussion in paragraph 1 and there is no paragraph 3.

Dear editor Omelchenko,

We would to resubmit our manuscript entitled “Overload wave-memory induces amnesia of a self-propelled particle”. We would like to thank all the referees for their constructive and supportive remarks.

We are very happy to see that the referee 2 and 3 both support publication. We have addressed the latest minor points of the referee 1 and we provide below a detailed answer to the final point raised by the referee 4. Our responses are marked in blue and are written in a point-by-point fashion below. The corresponding changed are marked in orange in the main manuscript.

We encountered a technical problem to insert the bib file. So, it has been uploaded as a supplementary dataset.

We hope very much our response will meet the requirements of referee 1 and 4.

All the best,

Matthieu Labousse, for the authors

Reviewer #1 (Remarks to the Author):

I would like to thank the authors for having considered my comments. The answers are clear and the improvement of the manuscript has definitely strengthened the work. From my side, the answers of the authors made clear the difference between this work and the previous ones based on the same techniques and recent studies in Active Matter. I think their improvements made the revised version of the manuscript suitable for Nat Comm. after a minor revision.

Minor points:

1. Maybe the question is silly, but I think it might naturally emerge for a reader that is not an expert in the specific field: Why the regime between low and infinite is called "high memory regime"? This definition makes infinite small than higher and thus sounds odd. I would ask the authors to explain better these three regimes.

We thank the referee for this crucial comment. The adjective "low", "intermediate", "high", and "infinite" used to refer to the different regimes relates to the value of the memory parameter Me .

- In the "Low" memory regime, with the smallest Me , the wavefield only provides propulsion and no organized trajectory is observed without external forces. Typically, this regime is such as $Me < 20$ for stiff harmonic confining potentials.
- In the "Intermediate" memory regime, the trajectory of the droplet self-organizes. Lemniscates, trifolids and quantified circular trajectories are observed there. Typically, this regime is such as $20 < Me < 135$ depending on the frequency of the external potential.
- In the "High" memory regime, as investigated throughout this article, the trajectories are chaotic as a result of a Shilnikov bifurcation. This is the regime of this paper, and $Me > 135$ typically leads to a fully developed chaotic behavior.
- The "Infinite" memory regime is obtained by removing the time decay of the waves. In such case, correlations within the dynamics are destroyed and the wavefield behaves like our randomly generated field.

To better explain these notions, we write differently the description of figure 1 (d,e,f) (see page2, section "Dynamics of the particle: overview of the regimes with the memory", 1st paragraph.)

2. Fig 1 In the caption the definition of τ_F might help.

Following the referee's comment, we added the definition in the caption of Figure 1.

3. Eq. (1): I found "higher-order term" too generic

The notation is indeed lacking precision and can be misleading. Instead, we wrote $\mathcal{O}(\|\vec{v}\| \left(t_N \right) \nabla \zeta(\vec{r}_N, t_N)^2)$. Equation (1) has been modified, and also the reference to this correction in the main text.

4. It is still not clear, at least to me, the link between τ_F and the control parameters of active particles, e.g. self-propulsion speed, persistence length, etc... For instance: is there any relationship between memory length and persistence length of the particle?

The question about the role of τ_F in the active properties of the walker is a difficult one as it is connected to multiple dynamics.

Shortly, the time τ_F is here used as a natural time scale, as it sets the time step in Eqs.(1,2) which separate two interactions with the wave field in between two bounces.

This constrains in observing the walking regime usually forces experimentalists to fix the frequency of oscillation of the bath as a function of the bath viscosity and presence of obstacles. As a result, changing τ_F to investigate the change of propulsion has never been systematically investigated in experiments.

In order to emit long-lasting standing waves, the walker must be trigger subcritical Faraday waves at the liquid surface by its successive impacts. In order to do so, the walker bouncing period **must** match the Faraday waves period (therefore τ_F sets the faraday frequency and bouncing frequency). While we assume numerically to have a perfectly inelastic ball bouncing on a stiff surface, the experimental reality is much, much more complex. The walking droplet regime is actually observed in a narrow region of the parameter space. More precisely, the combination of droplet viscosity, liquid bath viscosity, frequency/amplitude of oscillation, and size of the droplet must hit a sweet spot where the droplet deformation is negligible between successive impacts, the surface deforms enough to have a significant waves emitted and the droplet bouncing frequency matches the Faraday frequency. (An extensive experimental investigation has been made in Wind-Willassen et al, Phys. of Fluids, 25:082002 (2013), and in Molacek and Bush, J. Fluid Mech., 727:612-647 (2012)).

For the more specific question of the referee about the parameters of active particles:

- The self-propulsion speed is related to the memory parameter through a complex formula (Oza et al, J. Fluid Mech., 737:552-570 (2013), specifically section 4.1 therein, where the authors provided an extensive discussion about this relation). In essence, the walking speed increases with the memory parameter at low Me and saturate for moderate and high values of Me (approximately above $Me = 20$).
- For the persistence length, we refer to the free-space walking case where the walker mimics a run-and-tumble dynamics (for an extensive discussion, please refer to Hubert et al, Phys. Rev. E, 100:032201 (2019)). In essence, the duration/length of straight-run phases decreases when the memory parameter increases, and eventually saturates for very large memory parameter. As a

result, one can conclude that the persistence length decreases with increasing memory parameter.

To account for the referee's question, we added a short discussion about the relation between memory and self-propulsion speed and persistence length on the manuscript (see page2, section "Dynamics of the particle: overview of the regimes with the memory", 1st paragraph.). We decided however to avoid the complex discussion about the influence of τ_F . Essentially, τ_F acts as a natural clock in the system.

Reviewer #2 (Remarks to the Author):

The authors have attended all my requests, so I have no further points to raise. Some aspects of the work escape my expertise, but I trust these are well covered by the other referees comments.

We would like to thank referee for his support and also for her/his constructive remarks in the previous round.

Reviewer #3 (Remarks to the Author):

The authors have made careful and comprehensive changes to the manuscript, amply satisfying my initial queries. I am now happy to recommendation publication of the manuscript in Nature Communications.

We would like to thank referee for his support and also for her/his constructive remarks in the previous round.

Reviewer #4 (Remarks to the Author):

The main question I have asked is a difficult one and the authors have made an attempt to address it. In the rebuttal the authors mention about the moderate number of dimensions (that can go up to 27 DOF for the wave field). And just below, that the main statistical consequence is from the excess of wave DOF. This is an example where the claims don't quite help. I find the reading still hard to follow, at least from my perspective.

In the Discussion, the authors have added (*): "As a result, the chaotic-like dynamics of the walker comes from the synergistic interactions between the particle and the wavefield, a feature which stems from the increasing memory parameters and amount of information." That the dynamics of the walker comes from the synergistic interactions between the particle and the wavefield, is the known vital ingredient of this wave-particle association. The amnesia-effect that stems from the increasing amount of information still puzzles me.

(*) In the rebuttal the authors write "Discussions about that point can be found in the "discussion" section, paragraph 3." I found some discussion in paragraph 1 and there is no paragraph 3.

We would like to apologize to the referee for the misleading numbering. There has been a typo in the reply to the first referee's report, and the discussion was indeed in the first paragraph of the "Discussion" section.

At the lowest memory parameter the system can be described by a nonlinear four-dimensional system (2 dimensions for the x-y position and 2 dimensions for the v_x - v_y for the speed) as shown by

[1] M. Labousse* S. Perrard*. Non-Hamiltonian features of a classical pilot-wave dynamics. Physical Review E 90 (2), 022913 (2014)

[2] Bush, J.W.M., Oza, A. and Molacek, J., 2014. The wave-induced added mass of walking droplets, J. Fluid Mech., 755, R7.

At low memory parameter, the memory effects can be encoded into an effective speed-dependent propulsion force [1]. At the next order of the development, there are an added mass effect [2] and a correction of the propulsive force. As a consequence, in this regime, the system can be mapped onto a four-dimensional set of equations. When the memory is further increased, this procedure is no longer accurate and the wave degree of freedom must be explicitly taken into account. The number of degrees of freedom, for instance at $Me=10000$ (Supplementary Fig S13g), goes up 31 (27 for the waves and 4 for the positions/speed) which is more than 7 times the number of dimensions of the system for low memory parameter, which we estimate as an important increase of the system's degrees of freedom. We have modified and specified accordingly the sentence suggested by the referee.

In the discussion section of the article, the previous sentences "As a result, the chaotic-like dynamics of the walker comes from the synergistic interactions between the particle and the wavefield, a feature which stems from the increasing memory parameters and amount of information." as been changed for "Therefore, the main statistical properties cannot result from a chaotic mixing of the wave modes but rather from an increase, up to a factor seven, of wave degrees of freedom (see Supplementary Fig S13g) which stems from the increasing memory parameters and amount of information. "